# HPV infection and the genital cytokine milieu in women at high risk of HIV acquisition

Lenine J.P. Liebenberg[1,2,8]*, Lyle R. McKinnon[1,2,3,8], Nonhlanhla Yende-Zuma[1,8], Nigel Garrett [1], Cheryl Baxter [1], Ayesha B.M. Kharsany [1,2], Derseree Archary[1,2], Anne Rositch[4], Natasha Samsunder[1], Leila E. Mansoor [1], Jo-Ann S. Passmore [1,5,6], Salim S. Abdool Karim[1,7] & Quarraisha Abdool Karim [1,7]

Human papillomavirus (HPV) infection correlates with higher rates of HIV acquisition, but the underlying biological mechanisms are unclear. Here we study associations between HPV and HIV acquisition and relate these to vaginal cytokine profiles in an observational cohort of women at high risk of HIV infection (CAPRISA 004, $n = 779$) and with 74% HPV prevalence. We report here that HPV infection associates with a 2.5-fold increase in HIV acquisition risk in this population (95% CI: 1.2–5.3). Among 48 vaginal cytokines profiled, cytokines associated with HPV infection overlap substantially with cytokines associated with HIV risk, but are distinct from those observed in HPV negative women. Although our data do not establish a causative link between HPV status and the risk of HIV, we suggest that increasing HPV vaccination coverage may carry an additional benefit of reducing the risk of contracting HIV infection, particularly in regions with high HPV prevalence.

[1] Centre for the AIDS Programme of Research in South Africa (CAPRISA), Durban, South Africa. [2] Department of Medical Microbiology, University of KwaZulu-Natal, Durban, South Africa. [3] Department of Medical Microbiology, University of Manitoba, Winnipeg, MB, Canada. [4] Johns Hopkins Bloomberg School of Public Health, Baltimore, MD, USA. [5] Institute of Infectious Disease and Molecular Medicine (IDM) and MRC-UCT Gynecological Cancer Research Centre, University of Cape Town, Cape Town, South Africa. [6] National Health Laboratory Service, Cape Town, South Africa. [7] Department of Epidemiology, Columbia University, New York City, NY, USA. [8] These authors contributed equally: Lenine J.P. Liebenberg, Lyle R. McKinnon, Nonhlanhla Yende-Zuma. *email: Lenine.Liebenberg@caprisa.org

Cervical cancer disproportionately affects women in southern Africa, with double and triple, respectively, the incidence and mortality rates that are observed worldwide[1,2]. Almost all cases of cervical cancer are caused by human papillomaviruses (HPV)[3]. These DNA viruses preferentially infect basal epithelial cells, and their effective evasion of host immunity can ultimately result in the formation of intraepithelial lesions and cervical cancer. Over 40 HPV types are known to infect the human genital tract, with 12–20 classified as being oncogenic or "high risk" for causing cervical cancer[4–6].

Two vaccine-preventable HPV types, HPV-16 and -18, contribute to over 70% of cervical cancer cases[7]. These types have been included in each of the three HPV vaccines showing high efficacy in preventing HPV infection[8–11]. Because the risk of HPV infection is known to be very high following sexual debut[12], efforts are underway to implement these vaccines in young women worldwide, as the degree of protection is much higher in HPV uninfected individuals.

Young women in sub-Saharan Africa bear a disproportionate burden of not only HPV[13], but also human immunodeficiency virus (HIV) infection[14–16]. In addition to its role as a major factor in the development of cervical and several anogenital cancers, HPV has also been associated with increased risk of HIV acquisition[17,18]. Multiple HPV strains can coinfect the same individual, thus life history of HPV infection can include simultaneous persistence, clearance (non-persistence), and/or acquisition of different HPV strains. Increased HIV infection risk in sub-Saharan Africa has been associated with prevalent HPV infection, infection with either oncogenic or non-oncogenic HPV types, infection with non-oncogenic HPV types, and concurrent infection with multiple HPV types[19–21]. Clearance of HPV infection occurs relatively quickly, with about half of infections clearing by 6 months; and more slowly over the next few years, with an average persistence of ~10 months[22]. In two studies that assessed HPV persistence and HIV risk empirically in sub-Saharan Africa, non-persistence was associated with increased HIV risk[19,21], suggesting that active immune responses to HPV might underlie the observed HPV–HIV associations.

Previous work by our group has demonstrated that elevated mucosal cytokines in the reproductive tract of HIV-uninfected women are strong predictors of subsequent HIV infection outcome[23,24]. However, the stimuli responsible for the elevated cytokine concentrations remain unclear. Here we hypothesized that the female genital immune environment may differ according to HPV infection status; a concept that, if confirmed, would shed light on possible mechanisms for the reported associations between HPV infection and HIV risk. It is biologically plausible for cervical HPV infection, or other potentially correlated changes in the female genital tract to impact HIV risk, since sexual transmission of HIV is primarily mediated through interaction of the virus with genital cellular targets for infection, in the same mucosa where HPV replicates. It is therefore reasonable that immune cell recruitment to eliminate or contain HPV infection in the genital epithelium may promote an immune environment that favors HIV infection. A precedent for this is supported by the requirement of an inflammatory response, particularly those mediated by T and NK cells, to effectively facilitate HPV-associated genital wart or CIN1 regression[25–30]. These activated NK cell correlates of HPV clearance are also biomarkers of increased HIV risk in CAPRISA 004 participants[31]; the T cells are relevant to HIV infection as preferred targets of infection (CD4+ T cells expressing the CCR5 co-receptor for HIV entry i.e. CD4 +CCR5+ T cells)[32–34]. Given that inflammatory cytokines in cervicovaginal lavage (CVL) fluid are strong surrogates of both absolute numbers of endocervical HIV target cells and impaired barrier function[35], we measure here multiple cytokines relevant to

cell recruitment (chemokines), regulation, barrier growth and repair, and mediation of adaptive and pro-inflammatory immune responses, and correlate these to HPV status over time.

Given the high HIV incidence that has been observed in KwaZulu-Natal, particularly in young women aged 15–24 years[14,36], we characterize the epidemiology and mucosal immunology of HPV–HIV relationships in this setting. Specifically, we leverage the existing data and specimens collected as part of the CAPRISA 004 effectiveness study of tenofovir gel, carried out in high-risk women from rural and urban KwaZulu-Natal[37]. These data provide insights into the epidemiology of HIV and warrant investigation on whether HPV vaccines can decrease the risk of HIV infection.

## Results

**Associations with prevalent HPV infection**. HPV DNA was detected in 73.8% (95% CI 70.7–76.9%) of participants at baseline (HPV+; Table 1). The factors associated with HPV status were younger age (mean ± SD 23.4 ± 4.8 vs. 25.5 ± 5.9, $p < 0.001$; Wilcoxon rank sum test), marital status (4.0% married vs. 11.8% unmarried, $p < 0.001$; Fisher's exact test), living away from the regular sex partner (89.5% HPV+ vs. 80.8% HPV−, $p < 0.001$; Fisher's exact test), and baseline infection with *Chlamydia trachomatis* (15.9% HPV+ vs. 4.3% HPV−, $p = 0.003$; Fisher's exact test; Table 1). No differences in HPV prevalence were evident on the basis of study arm (tenofovir versus placebo), and HPV prevalence did not differ between age at sexual debut, number of lifetime partners, HSV-2 status, self-report of STI symptoms, condom use, sexual activity, or other behavioral parameters. The majority of the population was using injectable hormonal contraception (647/799; 84%), and no association between HPV status and contraception use was observed ($p = 0.274$; Fisher's exact test).

**Type-specific HPV prevalence**. About half of participants were infected with an oncogenic HPV type (51.9%, 404/779), and baseline coverage of vaccine types was 16.8%, 22.0%, and 41.3% for the Cervarix®, Gardasil®, and Gardasil®−9 vaccines, respectively. The most prevalent HPV types were the oncogenic HPV 16 (10.8%), 51 (9.8%), and 35 (9.4%), and the non-oncogenic viruses HPV 84 (11.7%) and 62 (9.6%). Approximately one-quarter of the cohort (24.3%) was infected by a single HPV type. The proportions of participants infected by 1–2, 3–5, >5 HPV types were 42.1%, 25.4%, and 6.3%, respectively, suggesting that multiple-type HPV infections are relatively common in this population.

**Impact of baseline HPV infection on HIV acquisition risk**. The 779 women in the study were followed-up for 1,213 person-years (py; mean of 18 months/participant), during which 66 HIV infections occurred (Table 2). HPV prevalence was associated with a 2.5-fold increased risk of HIV acquisition in multivariable analysis (aHR 2.5, 95% CI 1.2–5.3, $p = 0.015$; Cox proportional hazards regression). Infection with oncogenic HPV was associated with a 2.8-fold increased HIV risk ($p = 0.008$; Cox proportional hazards regression; Table 2). HIV risk was increased more than 2-fold if women were infected with HPV types preventable by the Gardasil® and Gardasil®-9 vaccines (Table 2; Supplementary Table 1). Infection with only non-Gardasil®-9 HPV types was associated with a 2-fold increase in HIV risk relative to HPV negative women, although not statistically significant (Table 2). Further, the HIV incidence rate (IR) increased dramatically with increasing numbers of infecting HPV types (IR 4.3, 95% CI: 2.7–6.5 for infection with 1–2 HPV types; IR 8.4, 95% CI: 5.5 to 12.5 for infection with 3–5 HPV types; IR 15.2,

## Table 1 Baseline characteristics

| | Overall (*n* = 779) | HPV+ (*n* = 575) | HPV− (*n* = 204) | *p*-value |
|---|---|---|---|---|
| *Demographics* | | | | |
| Age, mean (SD) | 24.0 (5.2) | 23.4 (4.8) | 25.5 (5.9) | <0.001 |
| Tenofovir group | 50.3 (392) | 50.6 (291) | 49.5 (101) | 0.807 |
| Income < R1000/month | 91.2 (634/695) | 90.1 (465/516) | 94.4 (169/179) | 0.092 |
| Married | 6.0 (47) | 4.0 (23) | 11.8 (24) | <0.001 |
| *Sexual behavior* | | | | |
| Sexual debut median (IQR) | 17 (16–19) | 17 (16–18) | 17 (16–19) | 0.746 |
| Lifetime partners | 2 (1–3) | 2 (1–3) | 2 (1–3) | 0.694 |
| Sex acts in last 30 days | 6 (4–10) | 6 (3–10) | 6 (4–12) | 0.069 |
| Living with regular partner | 12.7 (98/772) | 10.5 (60/574) | 19.2 (38/198) | <0.001 |
| Work away from regular partner | 24.0 (185/772) | 23.5 (135/574) | 25.3 (50/198) | 0.003 |
| Knows partner has other partners | 21.1 (161/763) | 21.1 (119/563) | 21.0 (42/200) | 0.113 |
| Partner consumes alcohol before sex | 29.7 (231) | 28.0 (161) | 34.3 (70) | 0.083 |
| Always used condoms | 71.1 (554) | 71.1 (409) | 71.1 (145) | 1.000 |
| Ever had sex for money | 1.9 (15) | 1.7 (10) | 2.5 (5) | 0.555 |
| *Biological* | | | | |
| STI symptoms at baseline | 37.2 (290) | 38.4 (221) | 33.8 (69) | 0.273 |
| HSV-2 antibodies | 52.3 (407/778) | 50.5 (290/574) | 57.4 (117/204) | 0.103 |
| Positive STI result[a] | 22.5 (60/267) | 25.2 (38/151) | 19.0 (22/116) | 0.241 |
| Neisseria gonorrhoeae | 0.4 (1/267) | 0.7 (1/151) | 0 (0/116) | 1.000 |
| Chlamydia trachomatis | 10.9 (29/267) | 15.9 (24/151) | 4.3 (5/116) | 0.003 |
| Trichomonas vaginalis | 9.4 (25/267) | 8.0 (12/151) | 11.2 (13/116) | 0.401 |
| Mycoplasma genitalium | 3.0 (8/267) | 4.0 (6/151) | 1.7 (2/116) | 0.472 |
| HSV-1 (PCR) | 0 (0/267) | 0 (0/151) | 0 (0/116) | — |
| HSV-2 (PCR) | 2.3 (6/267) | 2.0 (3/151) | 2.6 (3/116) | 1.000 |
| Haemophilus ducreyi | 0 (0/267) | 0 (0/151) | 0 (0/116) | — |
| Treponema pallidum | 0 (0/267) | 0 (0/151) | 0 (0/116) | — |
| C. trachomatis LGV strains | 0 (0/267) | 0 (0/151) | 0 (0/116) | — |

*HPV+ cervicovaginal HPV DNA detected, HPV- no cervicovaginal HPV DNA detected, SD standard deviation, IQR interquartile range, LGV lymphogranuloma venereum, PCR polymerase chain reaction*
[a]Diagnosis based on the multiplex polymerase chain reaction (PCR)

## Table 2 Hazard ratio for HIV acquisition by HPV status

| HPV Status | Person years (N) | # HIV events | HIV IR/100py (95% CI) | Hazard Ratio[a] (95% CI) | *p*-value |
|---|---|---|---|---|---|
| *Baseline* | | | | | |
| Baseline HPV Negative (reference) | 331.4 (204) | 8 | 2.4 (1.0–4.8) | — | — |
| Prevalent HPV | 882.1 (575) | 58 | 6.6 (5.0–8.5) | 2.5 (1.2–5.3) | 0.015 |
| # prevalent types < 3 | 513.1 (328) | 22 | 4.3 (2.7–6.5) | 1.7 (0.8–3.9) | 0.201 |
| # prevalent types ≥ 3 | 369.0 (247) | 36 | 9.8 (6.8–13.5) | 3.8 (1.7–8.2) | <0.001 |
| Prevalent Oncogenic types | 625.5 (404) | 46 | 7.4 (5.4–9.8) | 2.8 (1.3–6.0) | 0.008 |
| Prevalent Non-oncogenic types | 256.6 (171) | 12 | 4.7 (2.4–8.2) | 1.9 (0.8–4.7) | 0.179 |
| Prevalent Cervarix® types | 203.2 (131) | 12 | 5.9 (3.1–10.3) | 2.0 (0.8–5.1) | 0.151 |
| Prevalent Gardasil® types | 261.7 (171) | 18 | 6.9 (4.1–10.9) | 2.6 (1.1–6.2) | 0.035 |
| Prevalent Gardasil®9 types | 495.9 (322) | 36 | 7.3 (5.1–10.0) | 2.7 (1.2–6.0) | 0.012 |
| Non-Gardasil®9 types | 386.2 (253) | 22 | 5.7 (3.6–8.6) | 2.2 (0.9–5.1) | 0.054 |
| *Longitudinal* | | | | | |
| Category 1: Remained HPV negative (reference) | 178.6 (108) | 2 | 1.1 (0.1–4.0) | — | — |
| Overlapping HPV Categories | | | | | |
| Acquired any HPV types | 633.1 (405) | 49 | 7.7 (5.7–10.2) | 9.2 (2.1–40.5) | 0.003 |
| Cleared any HPV types | 747.3 (477) | 54 | 7.2 (5.4–9.4) | 6.3 (1.5–26.3) | 0.012 |
| Persisted any HPV types | 358.7 (236) | 28 | 7.8 (5.2–11.3) | 6.4 (1.5–27.4) | 0.013 |
| Mutually Exclusive HPV Categories | | | | | |
| Category 2: Only Cleared | 218.5 (130) | 8 | 3.7 (1.6–7.2) | 2.2 (0.4–11.0) | 0.333 |
| Category 3: Only Acquired | 141.6 (85) | 6 | 4.2 (1.6–9.2) | 5.5 (0.9–33.6) | 0.068 |
| Category 4: Only Persisted | 52.5 (33) | 1 | 1.9 (0–10.6) | 0.8 (0.0–21.4) | 0.888 |
| Category 5: Cleared and Persisted | 90 (61) | 6 | 6.7 (2.4–14.5) | 6.4 (1.3–32.2) | 0.024 |
| Category 6: Cleared and Acquired | 275.3 (178) | 22 | 8.0 (5.0–12.1) | 7.8 (1.8–34.3) | 0.006 |
| Category 7: Persisted and Acquired | 52.8 (34) | 3 | 5.7 (1.2–16.6) | 5.6 (0.9–34.5) | 0.061 |
| Category 8: Cleared, Acquired, and Persisted | 163.4 (108) | 18 | 11 (6.5–17.4) | 10.2 (2.3–45.6) | 0.002 |

[a]Adjusted for the following baseline covariates: treatment group, age, age of sexual debut, number of sexual partners, number of sex acts (past 30 days), presence of STI, HSV-2 status, condom use, marital status, and whether women were living with regular partners. CI, confidence interval

**Table 3 HIV risk by the number of infecting HPV types**

|  | Number of People | Person-years | Number of HIV events | Crude incidence rate | Lower 95% bound | Upper 95% bound | *p*-value for trend |
|---|---|---|---|---|---|---|---|
| PREVALENCE |  |  |  |  |  |  |  |
| 1–2 types | 328 | 513.1 | 22 | 4.3 | 2.7 | 6.5 | <0.001 |
| 3–5 types | 198 | 296.4 | 25 | 8.4 | 5.5 | 12.5 |  |
| >5 types | 49 | 72.6 | 11 | 15.2 | 7.6 | 27.1 |  |
| INCIDENCE |  |  |  |  |  |  |  |
| 1–2 types | 165 | 254.8 | 18 | 7.1 | 4.2 | 11.2 | 0.042 |
| 3–5 types | 126 | 193 | 17 | 8.8 | 5.1 | 14.1 | . |
| >5 types | 29 | 43.8 | 8 | 18.3 | 7.9 | 36 |  |
| CLEARANCE |  |  |  |  |  |  |  |
| 1–2 types | 249 | 397.5 | 18 | 4.5 | 2.7 | 7.2 | 0.001 |
| 3–5 types | 183 | 281.3 | 26 | 9.2 | 6 | 13.5 |  |
| >5 types | 45 | 68.5 | 10 | 14.6 | 7 | 26.8 |  |
| PERSISTENCE |  |  |  |  |  |  |  |
| 1–2 types | 101 | 158.5 | 5 | 3.2 | 1 | 7.4 | 0.002 |
| 3–5 types | 98 | 142.7 | 14 | 9.8 | 5.4 | 16.5 | . |
| >5 types | 37 | 57.4 | 9 | 15.7 | 7.2 | 29.8 | . |

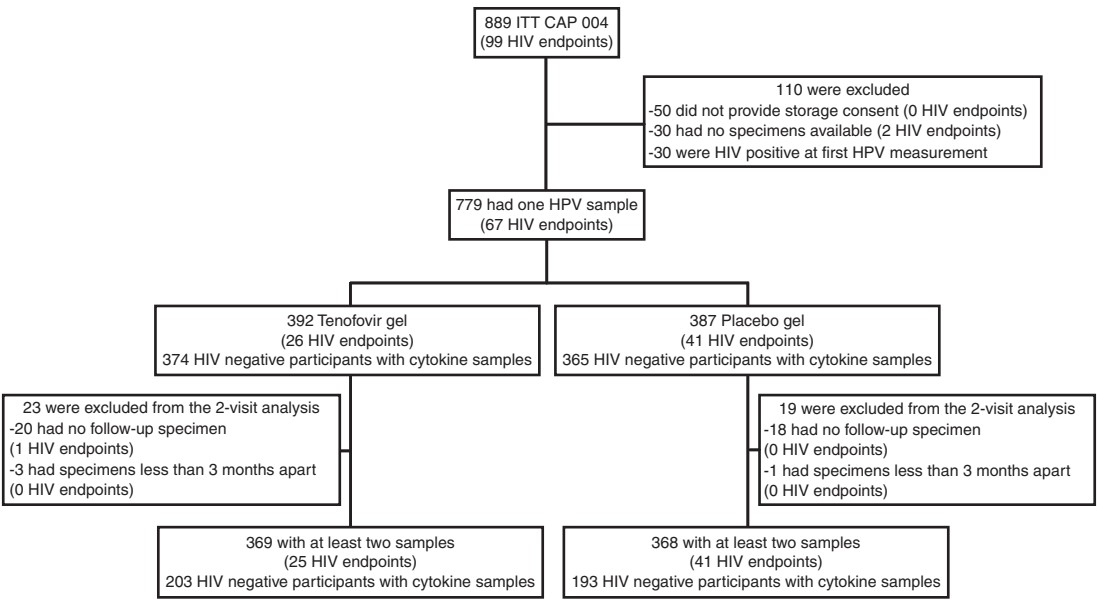

**Fig. 1** Enrollment and follow-up of study participants

95% CI: 7.6–27.1 for infection with more than 5 HPV types; *p* < 0.001; Log-rank test for trend; Table 3).

**Increased HIV risk with cleared, acquired and/or persistent HPV**. We next assessed whether a change in HPV status between visits had an impact on HIV acquisition rates [*N* = 737; median 16 months in the study (IQR: 12–19); Table 2; Fig. 1]. Three overlapping HPV classifications were defined as follows: women who "acquired any" HPV types, who "cleared any" HPV types, and who "persisted with any" HPV types. Each category was defined irrespective of any other type-specific changes occurring between the two study visits (Table 2; Fig. 2). As such, a woman who had evidence of acquiring, clearing and persisting any HPV types between visits would have contributed to the numbers observed in each of the three groups. The term "cleared" is synonymous here with the term "non-persistence", and refers to the presence of DNA for an HPV type at one visit but not at the next consecutive visit, acknowledging the challenges of determining whether this

represents true elimination or latent HPV infection[38]. Similarly, the term "acquired" here refers to the absence of DNA for an HPV type at one visit but the detection of it at the next consecutive visit, acknowledging that this definition may reflect new or re-infection, or reactivation[39]. Clearance of any HPV type was observed in most participants (477/737; 64.7%; Fig. 2). Compared to those who remained HPV negative, in adjusted analyses, increased HIV risk was observed for acquiring any HPV type (aHR 9.2, 95% CI 2.1–40.5, *p* = 0.003; Cox proportional hazards regression), clearing any HPV type (aHR 6.3, 95% CI 1.5–26.3, *p* = 0.012; Cox proportional hazards regression), and for persistence of any HPV type (aHR 6.4, 95% CI 1.5–27.4, *p* = 0.013; Cox proportional hazards regression; Table 2; Fig. 3; Supplementary Table 2). Similar to our HPV prevalence analysis, HIV risk increased with the number of HPV types that were acquired, cleared and/or that persisted (Table 2). The use of 1% tenofovir gel during the CAPRISA 004 trial[37,40] neither prevented HPV infection nor affected its persistence or clearance (Supplementary Table 3).

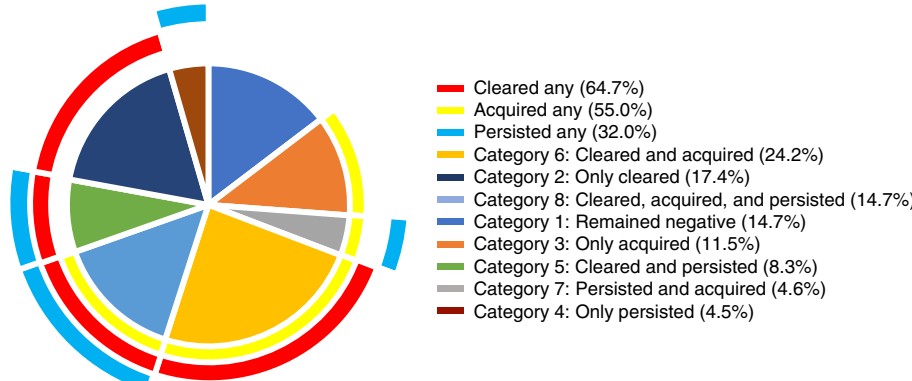

Cleared any (64.7%)
Acquired any (55.0%)
Persisted any (32.0%)
Category 6: Cleared and acquired (24.2%)
Category 2: Only cleared (17.4%)
Category 8: Cleared, acquired, and persisted (14.7%)
Category 1: Remained negative (14.7%)
Category 3: Only acquired (11.5%)
Category 5: Cleared and persisted (8.3%)
Category 7: Persisted and acquired (4.6%)
Category 4: Only persisted (4.5%)

**Fig. 2** The number of participants per HPV category in the CAPRISA 004 cohort ($N = 737$). HPV outcomes were classified into three overlapping categories where any HPV type could have simultaneously been acquired (Acquired Any HPV), cleared (i.e. eliminated or kept latent; Cleared Any HPV), or persisted (Persisted Any HPV). Additionally, eight mutually exclusive categories were defined: [Category 1] women who remained HPV negative ($n = 108$), or converted [Category 2] from HPV positive to HPV negative status (*only cleared*; $n = 130$) between visits, [Category 3] from HPV negative to HPV positive (*only acquired*; $n = 85$), or [Category 4] only retained the same HPV types between visits (*only persisted*; $n = 33$); or women who had a more mixed natural history whereby they [Category 5] cleared some HPV types while others persisted ($n = 61$), [Category 6] cleared some and acquired new types ($n = 178$), [Category 7] acquired some and some persisted ($n = 34$), or [Category 8] cleared some, acquired some, and some persisted ($n = 108$). Percentages reflect the frequency of observations among overlapping definitions ($N = 737$), and mutually exclusive definitions ($N = 737$)

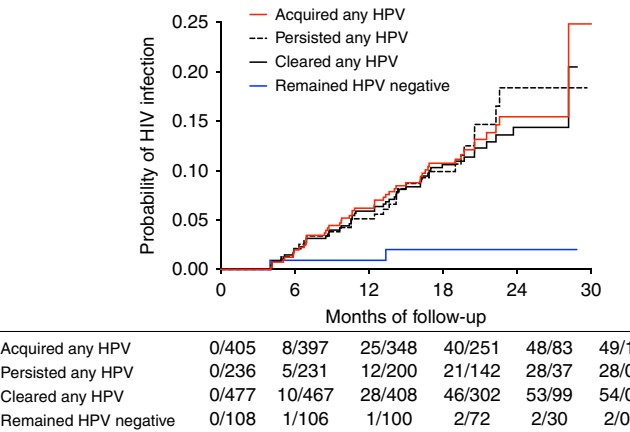

| | | | | | | |
|---|---|---|---|---|---|---|
| Acquired any HPV | 0/405 | 8/397 | 25/348 | 40/251 | 48/83 | 49/1 |
| Persisted any HPV | 0/236 | 5/231 | 12/200 | 21/142 | 28/37 | 28/0 |
| Cleared any HPV | 0/477 | 10/467 | 28/408 | 46/302 | 53/99 | 54/0 |
| Remained HPV negative | 0/108 | 1/106 | 1/100 | 2/72 | 2/30 | 2/0 |

**Fig. 3** Kaplan–Meier curves representing the relationship between HPV infection and HIV risk. Associations of HIV incidence were determined by modeling time to HIV infection with multivariable Cox proportional hazards regression models, where time spent in the study was calculated from randomization to the estimated date of HIV infection or the date of withdrawal from the study

**HPV clearance is most consistently associated with HIV risk.** To circumvent uncertainty in the extent of overlap between the above-mentioned HPV groups, HPV outcomes were next classified into eight mutually exclusive categories to further probe the impact of HPV on HIV risk and cytokines. The first four categories defined women with more straightforward changes in HPV status over time: Category 1, women who remained HPV negative (i.e. HPV DNA negative at both visits); Category 2, women who only cleared HPV (i.e. HPV DNA detected at the first visit, but not the next); Category 3, women who only acquired HPV (i.e. HPV DNA negative at the first visit, but positive at the next); and Category 4, women with no changes to the HPV types detected at both visits. The remaining categories captured women who were HPV DNA positive at both visits, with combinations of HPV acquisition, clearance and/or persistence. These women were classified as Category 5, women who both cleared and retained some HPV types; Category 6, women who both cleared and

acquired some HPV types; Category 7, women who persisted and acquired some HPV types; and Category 8, women with evidence of clearing, acquiring, and persisting with some HPV types (Table 2; Fig. 2). In multivariable analyses, increased HIV risk was observed in all HPV infection categories compared to women who remained HPV negative; significantly so for those in categories 5–8 (Table 2). Of these, the only category not significantly associated with HIV risk was the one lacking observations of clearance (Category 7; persisted and acquired; Table 2). These data suggest that the high frequency of HPV clearance in the cohort underscores its large contribution to increased rates of HIV acquisition, and support the hypothesis that effective immune responses against HPV infection may contribute to HIV risk[19,21,39].

**Association between prevalent HPV infection and genital cytokines.** Given the associations between HPV and HIV observed in this study, we hypothesized that the mucosal cytokine milieu may vary by HPV status. To test this, we compared the concentrations of cytokines in CVL fluid[24] of women with and without evidence of HPV at baseline. Multivariable linear regression models were used to determine the associations between baseline HPV status and the concentrations of 48 cytokines in 740/779 women with both cytokine and HPV data at baseline, adjusting for covariates relevant to HPV infection and HIV risk. Consistent with the HPV–HIV associations, infection with any HPV type at baseline (Fig. 4a) was associated with increases in several cytokines relative to the HPV negative group. Cytokines previously associated with HIV risk in CAPRISA 004 participants[41] (IL-8, MIP-1α, and MIP-β) were elevated in the genital tract of HPV positive women relative to HPV negative women (Fig. 4). A similar profile of elevated cytokine concentrations was also observed with infection by any of the nine vaccine-preventable HPV types relative to HPV negative women (Fig. 4b). Further, although IL-8, IP-10, MIP-1α, MIP-β were each significantly elevated relative to the HPV negative group, concentrations of all but IP-10 remained significantly so after adjustment for multiple comparisons, with IP-10 exhibiting borderline statistical significance ($p = 0.063$; Linear regression; Fig. 4b).

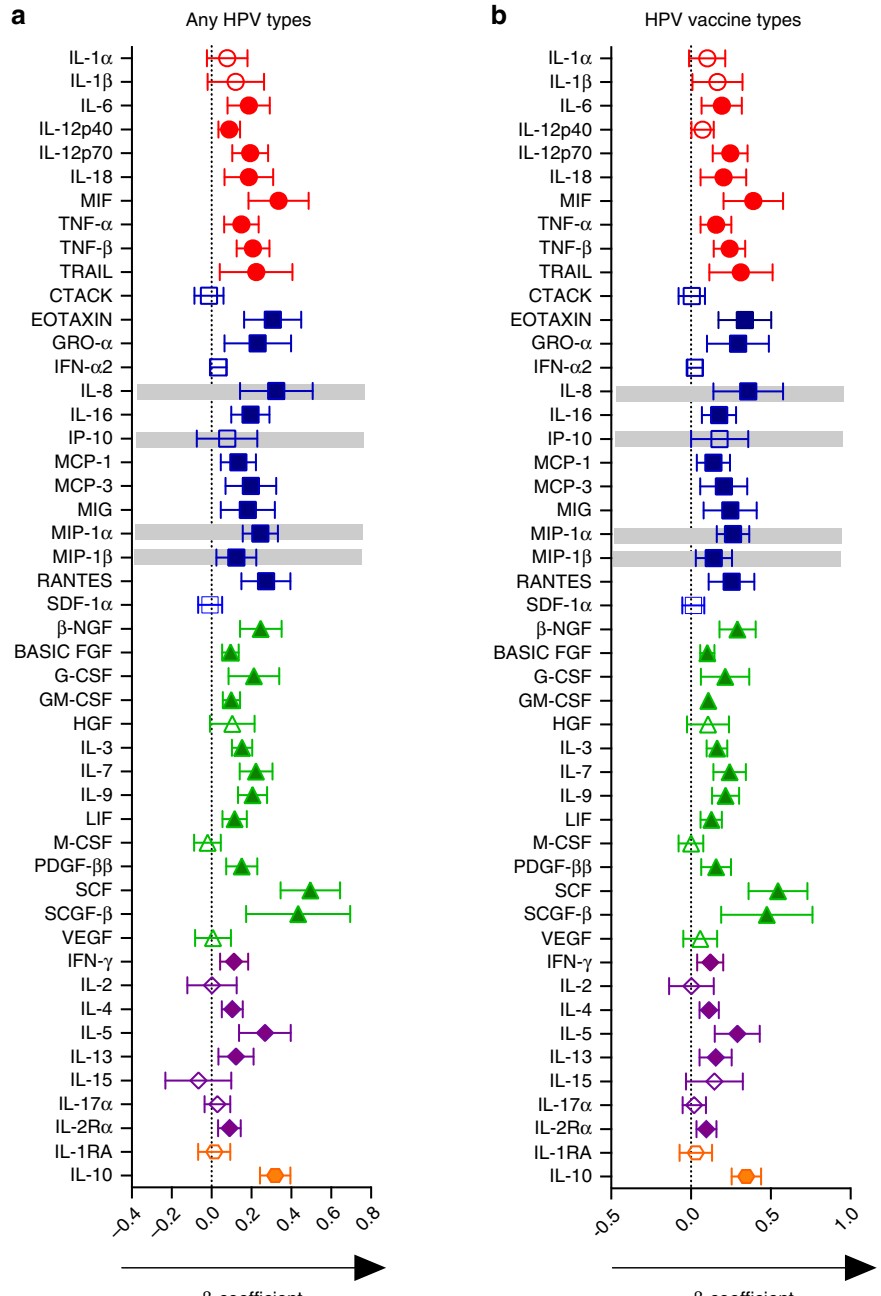

**Fig. 4** Association between prevalent HPV infection and genital cytokines. Plots depict the mean differences in genital cytokine concentrations between women with detectable HPV DNA and those without (N = 740; Fig. 4a), and specifically in women infected with any of the nine preventable HPV vaccine types compared to women with no evidence of genital HPV DNA (N = 472; Fig. 4b). Individual associations are shown between HPV status and pro-inflammatory cytokines (red), chemokines (blue), growth/hematopoietic factors (green), adaptive response cytokines (purple), and regulatory cytokines (orange), with error bars depicting 95% confidence intervals. The models test the hypothesis that the mean cytokine levels in comparator groups are equal i.e. β = 0. The dotted line at β = 0 distinguishes higher (to the right of the line) from lower (to the left of the line) mean cytokine differences in the respective categories relative to the HPV negative group. Gray shadings represent genital cytokines previously associated with HIV risk in this cohort: IP-10, IL-8, MIP-1α, and MIP-1β. Filled shapes represent significant associations after false discovery rate (FDR) adjustment using a threshold of 0.05 in multivariable models controlling for baseline treatment group, age, age of sexual debut, number of sexual partners number of sex acts (past 30 days), presence of STI symptoms, HSV-2 status, condom use, marital status, and whether women were living with regular partners

**Genital tract cytokines by HPV status over time**. We next assessed using linear mixed models whether prospectively defined HPV phenotypes, which all showed strong HIV risk associations, were also associated with differences in genital cytokine concentrations (Fig. 5). Although the number of genital sampling visits in the trial ranged from 1 to 4, the majority of HIV-negative specimens were available for 2, approximately 9 months apart.

Similar to the baseline analysis, significant increases in genital cytokine concentrations were observed at visits where the overlapping definitions of HPV clearance (N = 482), acquisition (N = 404) or persistence (N = 247) were observed, respectively, relative to remaining HPV negative (N = 173; Fig. 5a). Among the three overlapping HPV categories, concentrations of only two cytokines were significantly reduced relative to those remaining HPV

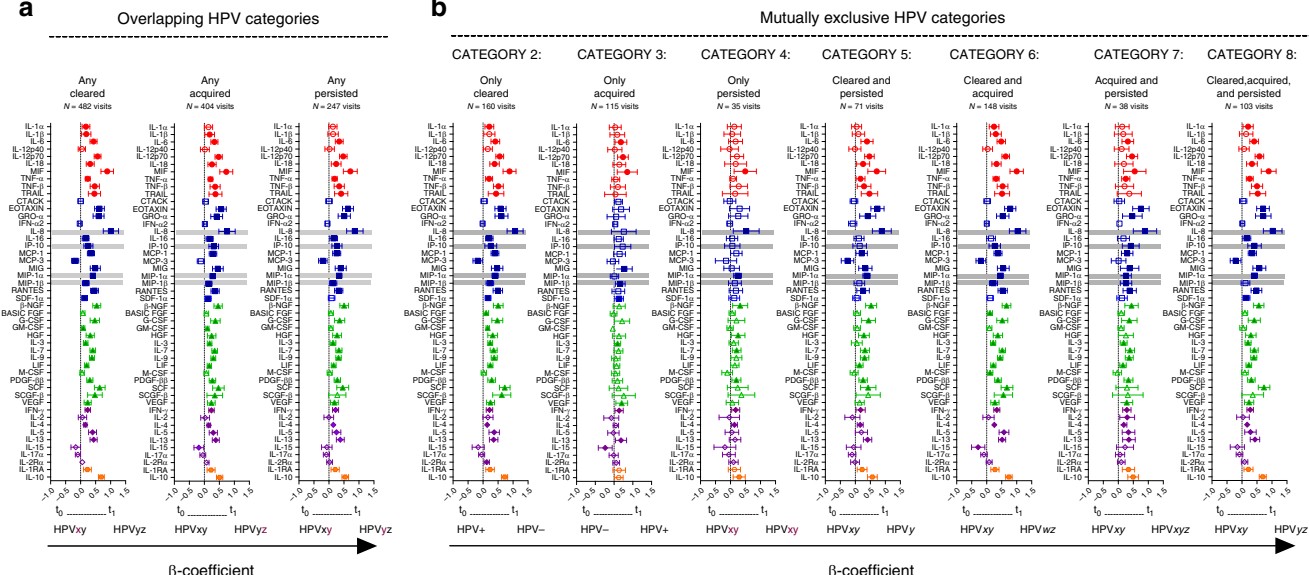

**Fig. 5** Association between HPV infection and genital cytokines over time. Figure 5a: Plots depict the mean differences in genital cytokine concentrations at visits where the overlapping definitions of HPV clearance ($N = 482$ visits), acquisition ($N = 404$ visits) and persistence ($N = 247$ visits) were observed, respectively, relative to category 1, where participants remained HPV negative ($N = 173$ visits). Figure 5b: Mean differences in genital cytokine concentrations between category 1 ($N = 173$) and visits where the HPV categories 2–8 were observed. The number of visits compared to category 1 in multivariable models are listed beneath each category. Individual associations are shown between HPV status and pro-inflammatory cytokines (red), chemokines (blue), growth/hematopoietic factors (green), adaptive response cytokines (purple), and regulatory cytokines (orange), with error bars depicting 95% confidence intervals. The models test the hypothesis that the mean cytokine levels in comparator groups are equal i.e. $\beta = 0$. The dotted line at $\beta = 0$ distinguishes higher (to the right of the line) from lower (to the left of the line) mean cytokine differences in the respective categories relative to the remained HPV negative group. Gray shadings represent genital cytokines previously associated with HIV risk in this cohort: IP-10, IL-8, MIP-1α, and MIP-1β. Filled shapes represent significant associations after false discovery rate (FDR) adjustment using a threshold of 0.05 in multivariable models controlling for baseline treatment group, age, age of sexual debut, number of sexual partners, number of sex acts (past 30 days), presence of STI symptoms, HSV-2 status, condom use, marital status, and whether women were living with regular partners. The figures are annotated to describe the temporal dynamics of the HPV infection, with $t_0$ referring to the HPV status at the last visit prior to that at $t_1$, at which cytokine concentrations were measured for both that category and for the category 1 control. Hypothetical HPV type infections are listed beneath to clarify the definition. For example: for "ANY HPV CLEARANCE" between t0 and t1, the clearance of HPV (HPV type-x) can be observed in the presence of acquisition (HPV type-z) or persistence (HPV type-y)

negative: IL-15 in women who acquired any HPV, and MCP-3 in women who cleared HPV or persisted with HPV infection. The cytokine responses were similarly broad among the categories, with significant associations observed for 39/48 (81%) cytokines, 38/48 (79%) cytokines and 34/48 (70%) cytokines in women who cleared, acquired or persisted with HPV infections, respectively, relative to those remaining HPV negative (Fig. 5a). Each category was also associated with increases in the four cytokines previously associated with HIV risk after adjustment for multiple comparisons (Fig. 5a).

We again applied the mutually exclusive HPV categories to better describe the specific contribution of each category to genital cytokine profiles (Fig. 5b). All significant associations observed were due to an elevation of cytokine concentrations relative to women who remained HPV negative; with the exception of MCP-3, which was significantly reduced in categories that included HPV clearance (Categories 2, 5, 6, and 8), and IL-15, significantly reduced in women who only acquired HPV (Category 3). HPV clearance alone (Category 2) had the most marked change in genital cytokine profiles; with 40/48 (83%) cytokines, 11/48 (23%) cytokines, and 10/48 (21%) cytokines significantly altered in women who only cleared, only acquired, and only persisted with HPV infections, respectively, relative to women remaining HPV negative (Fig. 5b). All categories had elevated concentrations of at least one of the cytokines known to be associated with HIV risk in CAPRISA 004

participants[41], with all four cytokines significantly increased in the categories where HPV was only cleared (Category 2), both cleared and acquired (Category 6), and where it was both acquired and persisted (Category 7; Fig. 5b).

**Distinct cytokine profile defined mainly by a history of HPV infection.** To better capture the multiple cytokines associated with HPV, we also used principal component analysis (PCA) to examine unsupervised multivariate clustering of cytokines by HPV group (Fig. 6). Women with a history of HPV infections segregated largely along principal component 1 (which explained 40% of the variance in cytokine expression), both supporting the regression analyses and emphasizing the general similarity in cytokine profile observed among women with a history of HPV infections relative to women who remained HPV negative. The model also identified a clustering of cytokines associated with remaining negative and only acquiring HPV, compared to all other mutually exclusive HPV categories; this was strongest by concentrations of IL-8, a cytokine significantly elevated in all categories but not in women who only acquired HPV (Fig. 5; Fig. 6).

## Discussion

This study confirmed the epidemiological link between HPV and increased risk of HIV infection, and tested the hypothesis that the female genital tract cytokine milieu would differ by HPV status.

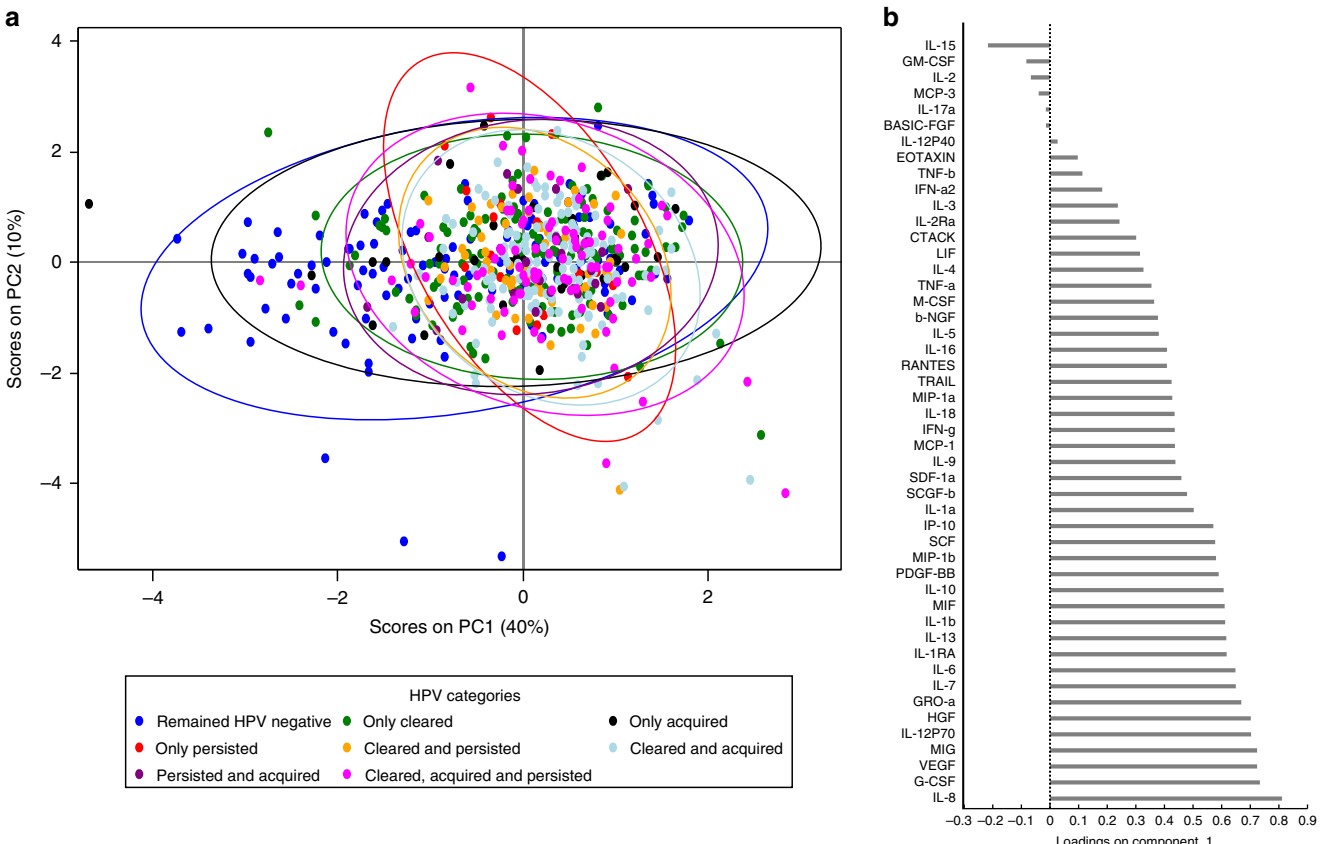

**Fig. 6** Principal component analysis (PCA) of the association between HPV infection and genital cytokines. (**a**) Principal component (PC) analysis and (**b**) cytokine loading on PC1 were used to define the relationship between HPV status and genital cytokine concentrations observed between two consecutive visits for all HIV-negative study participants ($N = 673$ visits). Each dot in the plot represents a participant's score, which indicates individual standing on each component, and different colors indicate the HPV category of each participant. Participants with higher scores in PC1 are expected to have a higher level of certain cytokines as those load higher in PC1; e.g. IL-8

The data presented are in support of this hypothesis, particularly in women who clear their HPV infection. Since clearance was most consistently associated with increased HIV risk, and with the broadest cytokine response, including all cytokines previously associated with HIV risk in this cohort[41], the data underscore the importance of further investigation to fully understand the complex immunological environment that may mediate the increased risk of HIV associated with HPV infection.

Approximately three-quarters of women in this HIV prevention trial of high-risk women were HPV-infected; this suggests a high overlap between burdens of both HPV and HIV infections in KwaZulu-Natal, South Africa. The prevalence of oncogenic HPV infections was 52%, somewhat lower than the 70% observed in a small female sex worker cohort in KwaZulu-Natal[42], but markedly higher than the 20.8% observed in Cape Town[43], where the HIV prevalence is the lowest in South Africa. The 2.5-fold increase in HIV infection observed for women with prevalent HPV in our study is in line with the HPV–HIV infection risk estimates published in the meta-analyses, which showed an approximate 2-fold effect[17,18]. Similarly, the estimates in our associations between HIV risk and HPV clearance and persistence resemble those observed in another study in sub-Saharan Africa that reported increased HIV risk when any HPV types were cleared, but no association when only type-specific persistence occurred[19]. This study is the first to investigate the impact of tenofovir microbicide gel on HPV infection in a clinical trial setting. Here, contrary to the observed impact on preventing HIV and HSV-2 infections[37,44],

the use of topical tenofovir microbicide gel was not associated with the prevention of HPV infection, nor its persistence or clearance in this region.

In addition, we and others[42,45] demonstrate a link between increased HIV risk and oncogenic HPV. These oncogenic types were predominantly represented by IARC class 1 HPV types and included several specific associations with vaccine-preventable HPV types. Considering the association between increased risk of HIV acquisition and infection by vaccine-preventable HPV types, these data underscore the importance of protecting young women from HPV infection through vaccination and cross-protection[46] against the broadest number of oncogenic HPV strains. The data also suggest the potential for increased HIV risk if infected by non-vaccine types, indicating that the risk to HIV transmission may not be related to the oncogenicity of the HPV alone, but to general immune control of HPV. This concept emphasizes the importance of larger studies of HIV incidence in regions with differential HPV vaccine uptake to comprehensively determine the excess HIV risk attributed to infection with vaccine-preventable versus non-preventable types. Our finding that infection with any of the 9 vaccine-preventable HPV types was associated with raised concentrations of several genital cytokines associated with HIV risk suggests that any HIV protection provided by HPV vaccination may be biologically mediated. Considering the observed overlap in cytokine profiles associated with HIV risk and with HPV clearance, persistence and acquisition, additional studies will be required to determine whether the ability of HPV vaccination to prevent HIV infection may need to

be bolstered through complementary efforts to address any immunological effects associated with infection by non-vaccine HPV types.

This study focused not only on genital immune environments associated with subsequent HPV clearance, persistence or acquisition (as have others[47–50]), but on associations between current HPV infection and female genital cytokines, including cytokine biomarkers of HIV risk. Our data suggest that prevalent HPV infection is associated with increased concentrations of several pro-inflammatory, chemotactic, regulatory, growth-related, and adaptive response-associated cytokines. While our study agrees with the increased concentrations of certain chemokines associated with prevalent HPV infection[49], our measurement of a broad and substantial panel of cytokines in a large clinical cohort allows a more robust and expanded assessment of cytokine associations with prevalent HPV. Importantly, many of the most prominent HPV-associated cytokines have previously been associated with HIV risk in this cohort and in other studies of African women[24,35,41,51]. Multivariable modeling showed heavy overlap in cytokine profile between mutually exclusive HPV categories, but reasonably strong delineation between HPV categories compared to those who remained HPV negative during the study. Considering the relationship between genital cytokines and HIV risk, these data highlight the importance of including HPV status in risk scoring profiles and emphasizes the benefit of prospective HPV measurements for informing potential HIV infection risk.

HPV clearance occurred frequently, was associated with increased HIV risk, and was the common thread that best captured increased cytokine concentrations compared to those who remained HPV negative. Of the eight categories, clearance alone was associated with the highest number of significant differences in single cytokine concentrations relative to controls. The contribution of transient HPV infections to the genital cytokine environment is therefore highly relevant and should likely be considered in biomedical efforts to reduce HIV infection. HPV infection can be resolved spontaneously, or through innate and adaptive immune responses. A potential mechanism for the relationship between inflammation and HIV risk is cytokine-mediated recruitment of HIV target cells to the genital tract. The strong association between HPV infection and genital cytokine responses suggests a role for cellular immunity in efforts to control HPV infection. Here, several cytokines, including beta chemokines critical for attracting $CCR5^+$ HIV target cells[33], and those defining T and NK cell responses implicated in HPV-associated wart regression (e.g. MCP-1, IL-8, IP-10)[26–30] were elevated in women with a history of HPV infection compared to HPV negative controls; more often significantly in categories featuring HPV clearance. However, while the association between HPV and increased genital cytokine concentrations is strong, the data cannot assign specific mechanisms of HPV-mediated HIV transmission, nor can it confirm a causal relationship between HPV-associated cytokines and HIV risk. Further studies are warranted to assess cellular and other mechanism(s) for the relationship between these viruses. Defining cellular sources of HPV-associated cytokines may also contribute to the design of targeted interventions to control the inflammation and reduce HIV risk.

Since HPV can be rapidly cleared, maintained and acquired[21,22,52], this study is limited by the potential for misclassification of HPV categories between visits, which may have contributed to the similar cytokine patterns observed between categories. The misclassification of HPV categories also extends to the difficulty in determining whether acquisition events represent reactivation or new infection, whether cleared infections refer to latent or eliminated infection, and the challenges of

understanding the immunological profiles associated with each. Considering that acute HIV infection is demonstrated to increase the number of HPV types detected[39], the survival analysis is limited by this potential for overestimation of the HIV incidence associated with multiple-type infection, or with clearance, persistence, or acquisition of HPV. Although the CAPRISA 004 trial included frequent HIV testing and rigorous retrospective viral load confirmations[37], HPV genotyping data were available for only baseline and study exit visits of most seroconverters, and longitudinal HPV classification would have been based on genotyping conducted after HIV infection. Additionally, the difficulty in timing the changes in HPV status in the context of HIV infection, we were unable to determine the degree of HPV–HIV co-infection occurring from the same infected partner. Cytokine data were, however, available for the majority of visits for each participant. Considering the established relationship between HIV infection and immune dysregulation, our linear mixed models only included cytokine data collected from confirmed HIV-negative samples with complementary HPV genotyping data for that visit, and therefore reflects the genital mucosal immune milieu associated with a current HPV status in HIV-negative women. Since cytokine data were available for confirmed HIV-negative specimens, we can only speculate on the immune profile associated with HIV infection that could promote an HPV status. Our data does, however, support the hypothesis that immune profiles observed in HPV infection may contribute in some way to the established link between HPV infection and HIV risk.

Another limitation of the study is that only a subset of clinically relevant HPV types was assessed in this study, meaning that some HPV negative women could have been infected by unmeasured types. However, had this been of significant consequence, the difference between cytokine concentrations and HIV risk in any HPV category relative to HPV negative women should not have been as strong. Data on cervical cytology were not documented within the CAPRISA 004 trial. However, considering the population was HIV uninfected and mainly in their twenties, we would expect very little high grade abnormal cytology. However, given that this was a microbicide safety trial, extensive vaginal observation data were available, and these data do not support any widespread clinically obvious abnormal cytology likely to confound the analysis presented. Lastly, this study was also limited in the number of participants with both HPV and vaginal microbiome data, and the relationship between HPV and the vaginal microbiome in this context is not well understood.

The strength of this study lies in its combination of very strong epidemiological data and mucosal immune assessments in a large prospective cohort, one in which HPV and HIV rates are extremely high. In summary, although not a definitive mechanism, these data provide some insight into how HPV might increase HIV risk and highlight the importance of further studies to validate and expand on the concepts raised. The data suggest, among others, an urgency in understanding and controlling the immune predictors of HPV status and multiple-type infection, in both HIV infected and uninfected women; in determining the cellular or other factors associated with the cytokines observed in the different HPV status categories; and in understanding the immune impact of HPV infection in vaccinated individuals and its relation to HIV risk. These data emphasize the importance of HPV vaccination since oncogenic HPV types are linked to HIV infection. The study assessed genital inflammation and HPV in specimens representing the mucosal surface over which sexual transmission of both viruses takes place, and demonstrated a difference in genital cytokines by HPV status over time, which informs possible mechanisms to explain the complex epidemiological relationship between HPV and increased HIV risk. While

these data do not rule out possible unmeasured explanations for the increased HIV risk associated with HPV infection, they do provide compelling biological evidence for elevated immune responses in HPV positive women, consistent with increased HIV infection risk. It is important to highlight the high prevalence of HPV (74%), and that the broad coverage of vaccine types observed in this study supports the notion that the current roll-out of HPV vaccination in South African schools could have a strong cancer prevention benefit. It is also important to note the frequent changes in HPV status detected in this cohort of young South African women (>50% acquired or cleared HPV infection) and each HPV category's link with HIV-associated cytokines. Combined with large effect estimates for increased HIV risk, the population attributable fraction of HIV acquisition due to HPV infection is substantial in this region with hyperendemic HIV prevalence. Therefore, in addition to preventing cervical cancer, further development of more effective ways to prevent and/or treat HPV infection, or modify its immunological effects, could have major HIV prevention benefits.

## Methods

**Study population and clinical procedures.** The CAPRISA 004 study was a Phase IIb clinical trial to evaluate the safety and efficacy of tenofovir 1% gel applied vaginally to prevent HIV infection. This trial was completed in 2010 and demonstrated both safety and an effectiveness of 39% in preventing HIV in the intent-to-treat analysis[37]. The enrollment criteria and characteristics of the population have been described[37,53], and details of those excluded can be found in Fig. 1. All women provided informed consent for participation in the trial, and participants were only included in this study if consenting to specimen storage for assessment of the trial's secondary objectives (Fig. 1). The trial (NCT00441298) and this secondary analysis was approved by the University of KwaZulu-Natal's Biomedical Research Ethics Committee (E111/06), Family Health International's Protection of Human Subjects Committee (#9946) and the South African Medicines Control Council (#20060835). At each visit, study staff administered a questionnaire that captured a range of demographic, clinical, reproductive and behavioral variables. HIV testing was conducted monthly using two rapid HIV antibody tests (Abbott determine and Unigold) and confirmed with two independent RNA-PCR assays at least one week apart. In addition, on detection of seroconversion RNA PCR was conducted on retrospective stored plasma specimens to identify the window period of HIV infection. The timing of HIV infection was defined as the midpoint between last negative and first positive tests (by PCR or rapid HIV antibody test); the date of seronegative infections was defined as 14 days before a positive PCR. The first genital specimen sampling of each participant was conducted a mean of $3.8 \pm 2.7$ months after randomization into study arm. Because data on genital cytokine concentrations and HPV infection were documented in this first genital specimen, these visits are referred to as "baseline" in this study. Genital specimens, including cervicovaginal fluid aspirates, cervicovaginal swabs, and CVL were stored at months 3, 12, 24 and at study exit, where exit could fall anytime between month 1 and month 30. While the number of study visits with genital sampling ranged from 1 to 4, most women had measurements for only 2 visits, months 3 and 12. Cytokine data were available for the majority of visits for each participant, and specimens available from confirmed HIV-uninfected genital mucosal sampling visits were included in the cytokine analyses. HPV genotyping data were only available for baseline and study exit for most participants, and longitudinal HPV classification may have been based on genotyping conducted after HIV infection for some participants. Genital specimens were not collected from menstruating participants, and specimens were not available for a small subset of women (Fig. 1). CVL pellets were used for HPV genotyping, genital swabs for STI testing, CVL supernatants for cytokine analysis[24], and cervicovaginal aspirates were used for measurements of Tenofovir concentrations by ultraperformance liquid chromatograph-mass spectrometry[37,40,54]. No assessments of genital epithelial and immune cell phenotype and function were possible as cervical cell sampling was not conducted during the trial.

**Collection of CVL specimens.** CVL specimens were collected by applying 3 ml of sterile saline to the ectocervix and aspirating the liquid from the posterior fornix. CVL specimens were transported on ice to the CAPRISA laboratory within 6 h, where samples were centrifuged and the CVL pellet and supernatant fractions were separated and each were stored in −85 °C for future use.

**Detection of HPV DNA and other STI.** DNA was extracted from CVL pellets using an automated MagNA pure instrument (Roche Diagnostics, Indianapolis, IN, USA) according to the manufacturer's instructions. HPV DNA was amplified by PCR using the Roche Linear Array® HPV Genotyping kit (Roche Diagnostics, Indianapolis, IN, USA) according to manufacturer instructions. Briefly, a pool of biotinylated primers was added to CVL pellet DNA to amplify HPV by defining

capture sequences of nucleotides within the polymorphic L1 region of the HPV genome. This assay distinguishes between 37 HPV genotypes, including those considered a significant risk factor for progression to cervical cancer (oncogenic HPV types). The HPV genotypes detected in this study include HPV 6, 11, 16, 18, 26, 31, 33, 35, 39, 40, 42, 45, 51, 52, 53, 54, 55, 56, 58, 59, 61, 62, 64, 66, 67, 68, 69, 70, 71, 72, 73, 81, 82, 83, 84, IS39, and CP6108; with CP6108 denoting a subgroup of HPV type 89, IS39 a subgroup of HPV type 82, HPV type 55 a subgroup of type 44, and HPV type 64 a subgroup of type 34. Further, the assay contains a cross-reactive probe hybridizing with HPV genotypes 33, 35, 52 and 58 from the Alpha 9 genus, and single probes for each type to aid in the determination of these similar HPV types. Additional primer pairs were included to amplify the human β-globin gene, thereby providing an internal control for the sample adequacy of the DNA extraction and amplification process.

**Assessment of other sexually transmitted infections.** Serology was performed for herpes simplex virus type 2 (HSV-2) using the HSV-2 gG2 enzyme-linked immunosorbent assay (Kalon Biologicals Ltd.). DNA was extracted from vulvo-vaginal swabs [Xtractor Gene (Corbett Robotics Pty Ltd.)] to detect the presence of other current sexually transmitted infections[55,56]. Herpes simplex virus types 1 and 2, *Treponema pallidum* and *Haemophilus ducreyi* and C. trachomatis lympho-granuloma venereum (LGV) strains were detected by real-time multiplex PCR (Rotor-Gene 3000 platform (Corbett Robotics Pty Ltd., Sydney, Australia), while *Chlamydia trachomatis*, *Neisseria gonorrhoeae*, *Trichomonas vaginalis*, *Mycoplasma genitalium*, were detected using in-house optimized real-time multiplex PCR tests (National Institute of Communicable Diseases, Johannesburg, South Africa)[55,56].

**Measurement of tenofovir concentrations in cervicovaginal fluid.** Tenofovir concentrations were measured in cervicovaginal fluid by ultraperformance liquid chromatograph-mass spectrometry[37,40,54]. The intraday and interday assay accuracy was 96–107% and 92–106%, respectively; both with a precision of ≤9%. The limit of quantitation for tenofovir in cervicovaginal fluid was 2 ng/mL. Samples below the limit of quantitation were reported as half the limit of quantitation, while samples below the limit of detection were reported as 0 ng/mL.

**Measurement of genital cytokine concentrations by multiplex ELISA.** The concentrations of cytokines were previously measured in undiluted CVL fluid of CAPRISA 004 participants using Luminex multiplex assays[24]. Here we report the measurements of 48 cytokines, including several pro-inflammatory, hematopoietic, regulatory, adaptive, and growth-related cytokines: interleukin (IL)-1β, IL-1Rα, IL-2, IL-4, IL-5, IL-6, IL-7, IL-8, IL-9, IL-10, IL-12p70, IL-12p40, IL-16, IL-18, IL-1A, IL-2RA, IL-3, IL-13, IL-15, IL-17, basic fibroblast growth factor (FGF), cutaneous T-cell attracting chemokine (CTACK), eotaxin, granulocyte colony-stimulating factor (G-CSF), granulocyte macrophage colony-stimulating factor (GM–CSF), growth regulated (GRO)-α, hepatocyte growth factor (HGF), interferon (IFN)-γ, IFN-α2, interferon gamma-induced protein (IP)−10, leukemia inhibitory factor (LIF), monocyte chemotactic protein (MCP)−1, MCP-3, macrophage colony-stimulating factor (M-CSF), monokine induced by gamma-Interferon (MIG), macrophage migration inhibitory factor (MIF), macrophage inflammatory protein (MIP)−1α, MIP-1β, nerve growth factor (NGF)-β, platelet derived growth factor (PDGF)-ββ, regulated upon activation normal T cell expressed and presumably secreted (RANTES), stem cell factor (SCF), stem Cell Growth Factor (SCGF)-β, stromal derived factor (SDF)−1α, tumor necrosis factor (TNF)−α, TNF-β, TNF-related apoptosis inducing ligand (TRAIL), and vascular endothelial growth factor (VEGF). Analyte concentrations were measured using Bio-Plex Pro Human Cytokine kits and a Bio-Plex Array Reader (Bio-Rad Laboratories) as previously described[24]. The sensitivity of these kits ranged between 0.2 and 45.2 pg/ml for each of the 48 cytokines measured. Data were collected using Bio-Plex Manager software (version 6), and a 5 PL regression formula was used to calculate sample concentrations from the standard curves. Cytokine levels below the lower limit of detection of the assay were reported as the midpoint between the lowest concentration measured for each cytokine and zero. All specimens were plated undiluted, and all laboratory assays were conducted blinded. Duplicates were included within plates to assess intra-plate variability ($n = 20$ duplicate wells per plate), and replicates were placed across separate plates to assess inter-plate variability ($n = 32$ replicate wells across plates). Cytokine assessments were repeated if correlation coefficients of associations between replicates were below Spearman rho = 0.8, and if significant differences in magnitudes among replicates were observed. Individual cytokines undetectable in >40% of specimens were assessed as categorical values. To minimize variability in longitudinal analyses all specimens from the same patient were placed in the same run.

**Statistical analyses.** Oncogenic HPV investigated here were defined as Group 1, 2 A carcinogens by the International Agency for Research[4] and correspond with several meta-analyses' definitions[17,18,21]: HPV types 16, 18, 31, 33, 35, 39, 45, 51, 52, 56, 58, 59, and 68. The remainder of the 37 types investigated are defined as low risk for cervical cancer. Several baseline HPV infection categories were defined in order to assess the relationship between HIV acquisition and HPV DNA detection (prevalent HPV infection), multitype HPV infection, oncogenicity, and with vaccine type HPV infection. Considering the high number of multitype HPV

infections, the potential for co-infection is acknowledged, particularly within the latter two groups. For example, while women in the non-Gardasil®9 HPV types group were infected exclusively with non-Gardasil®9 HPV types, women in the Gardasil®9 group could be infected exclusively with Gardasil®9 HPV types (27%), or co-infected with vaccine and non-Gardasil®9 HPV types (73%).

Three overlapping HPV classifications were defined as follows: women who (i) "acquired any" HPV types, irrespective of whether they concurrently cleared and/or had persistent HPV infections; (ii) who "Cleared any" HPV types, irrespective of whether they concurrently acquired and/or had persistent HPV infections; or (iii) "Persisted any" HPV types, irrespective of whether they concurrently acquired and/or cleared any HPV infections between any two study visits (Fig. 2). Eight mutually exclusive categories were also defined to better determine the contribution of clearance, persistence and acquisition on HIV risk and cytokines (Fig. 2). Categories 1–4 reflect changes in being positive for any HPV (HPV+) or negative for all types (HPV-) between visits. Category 1, HPV- at one visit and still HPV- at the next refers to women who "Remained Negative"; Category 2, HPV+ at one visit and HPV- at the next refers to women who "Only Cleared" HPV; Category 3, HPV- at one visit and HPV+ at the next refers to women who "Only Acquired HPV; and Category 4, HPV+ at both visits but infected with no more/no less than the same HPV types at both visits refers to women who "Only Persisted". The remaining categories describe populations of women classified as HPV+ at both visits in terms of the type-specific changes observed. Category 5 refers to women who both lost and retained HPV types; Category 6, women who both lost and gained HPV types; Category 7, women who retained and gained HPV types; and Category 8, women with evidence of losing, gaining, and retaining HPV types between visits.

Statistical models excluded the potential contribution of asymptomatic STIs and concurrent agents of bacterial vaginosis on genital cytokines as this data were not available for the majority of visits. However, in the subset of participants with laboratory measures of STIs available ($n = 494$), STIs were less prevalent (24%) than HPV, suggesting that concurrent STI is unlikely to fully explain the HPV-associated cytokine differences. Sensitivity analyses controlling for laboratory STI data were carried out in the "acquired any" HPV category (the only category with sufficient observations for reliable assessment; Supplementary Figure 1). HPV-cytokine estimates and significant associations were still similar whether laboratory STI data were included in the model or not; justifying the appropriateness of models including STI symptoms in the absence of laboratory STI data.

Categorical variables were compared using Fisher's exact test and continuous variables were compared using Wilcoxon rank sum test. Multivariable models test the hypothesis that the mean cytokine levels in comparator groups are equal, and all were adjusted for the following baseline covariates: treatment group, age, age of sexual debut, number of sexual partners, number of sex acts (past 30 days), presence of STI symptoms, HSV-2 status, frequency of condom use, marital status, and whether women were living with regular partners. Multivariable linear regression models were used to analyze the effect of baseline HPV status on the levels of each cytokine, controlling for false discovery rate (FDR) using the Benjamin–Hochberg test. Multivariable linear mixed models with a random effect of the individual participant were used to analyze the effect of HPV strata on the levels of each cytokine, controlling for FDR using the Benjamin–Hochberg test. Autoregressive of order 1 (AR(1)) structure was used to account for correlation between measurements from the same participant. This structure was well suited for our data since we have evenly spaced cytokines measurements. The log-rank test was used to test for trend in HIV IRs across ordinal variables. Ninety-five percent confidence intervals for HIV IRs were calculated based on the Poisson distribution. Associations of HIV incidence were determined by modeling time to HIV infection with multivariable Cox proportional hazards regression models, where time spent in the study was calculated from randomization to the estimated date of HIV infection or the date of withdrawal from the study. Due to the fact that cytokines are correlated, we used PCA to model the relationship between HPV status and genital cytokine concentrations observed at the study end visit of all HIV- study participants. The aim was to identify patterns of cytokine expression for different HPV categories while using information from all 48 cytokines together. All analyses were performed using SAS version 9.4 (Statistical Analysis Software, North Carolina, USA). Two-sided P-values less than 0.05 were considered statistically significant.

**Life sciences reporting summary**. Further information on experimental design and reagents is available in the Life Sciences Reporting Summary.

**Reporting summary**. Further information on research design is available in the Nature Research Reporting Summary linked to this article.

## Data availability

The source data underlying Figs. 2, 4, 5, 6, and Supplementary Figure 1 are provided as a Source Data file. Any other data will be made available through request on a dedicated portal on the CAPRISA website (https://www.caprisa.org/Pages/CAPRISAStudies).

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

## Acknowledgements

The authors would like to thank David Lewis and Venessa Maseko for their contribution to the STI testing; Kerry Hyland for statistical support; and Willard Cates Jr and Amanda Troxler at Family Health International, Durham, North Carolina, USA, for important discussions regarding these data. The study was funded by the USAID, implemented by FHI360 (0437.0196/805291), the National Institutes of Health (5R01AI111936 to J.S.P.), and the DST-NRF Centre of Excellence in HIV Prevention at CAPRISA. The original CAPRISA 004 1% tenofovir gel trial was funded principally by the United States Agency for International Development (USAID) through FHI360 and CONRAD with additional support provided by the South African Department of Science and Technology (DST). We would like to thank all study participants and CAPRISA staff for making the CAPRISA 004 trial possible. L.J.P.L. is funded by the South African National Research Foundation (NRF) Research Career Advancement Fellowship award, SANTHE Path to Independence award, and by a FLAIR Fellowship supported by the African Academy of Sciences and the Royal Society. L.R.M. is supported by a Canadian Institutes of Health Research (CIHR) New Investigator Award.

## Author contributions

L.J.P.L., N.G., L.E.M., A.B.M.K., Q.A.K., S.S.A.K. and J.S.P designed the study. L.J.P.L. performed the experiments. L.J.P.L., L.R.M. and N.Y.-Z. analyzed the data. L.J.P.L., L.R.M., N.Y.-Z., D.A., C.B., A.F.R., N.S., A.B.M.K., L.E.M., Q.A.K., S.S.A.K. and J.S.P. wrote the manuscript.

## Competing interests

The authors declare no competing interests as defined by Nature Research, or other interests that might be perceived to influence the results and/or discussion reported in this paper.

## Additional information

**Supplementary information** is avaliable for this paper at https://doi.org/10.1038/s41467-019-13089-2.

