## [Peer Review File · Nature Communications]

Reviewers' comments:

Reviewer #4, expert in biostatistics (Remarks to the Author):

No further comments. The authors have adequately responded to my critique.

Reviewer #5, expert in HPV epidemiology (Remarks to the Author):

This is an important descriptive study which lends important biological marker data to help generate specific mechanistic hypotheses for the consistent epidemiologic associations between HPV infection and increased HIV infection risk.

The authors have generally responded well to the comments of the reviewers. In my view, the disconnect between the reviewers and the authors is the fundamental tension between how far epidemiologic research can contribute to demonstration of causality when compared with experimental laboratory studies, remaining cognizant that the laboratory evidence may be so highly contrived that the extrapolation of results to a complex human biological system may also be limited.

To that end, in this reviewer's opinion, the study was well-designed epidemiologically and used state-of-the-art biomarker measures both for HIV, HPV, and cytokines. These are all commonly used in epidemiologic research.

It is suggested that the authors carefully review the manuscript for all causal inference statements, and tone them down to better reflect the observational nature of what is really a hypothesis-generating dataset (albeit a very important one). A few possible (non-exhaustive) examples:

1. Title could be changed to: "Differences in genital cytokine milieu by HPV infection status in women at high risk of HIV acquisition". This title does not implicitly assume that the differences are triggered by HPV infection, but may be correlatively associated with other changes in the female genital tract (FGT). There are valid hypotheses to suggest this as outlined in Nowak et al (JID 2011;203(8):1182-91) and Averbach (as referenced by the authors)

2. Discussion first sentence: "This study tested the hypothesis that HPV can increase HIV acquisition risk by modulating the female genital mucosa in a way that may facilitate HIV infection". In fact, this is not what the study tested. The study tested the hypothesis that the FGT cytokine milieu would differ by HPV status, and the nature of the difference could lead to more refined hypotheses regarding a biological mechanism to explain the increased risk of HIV observed in HPV positive women (especially those with clearance).

3. Discussion last paragraph line 355 (clean): add the following (CAPS) to sentence: "In summary, we provide here DATA HELPFUL IN THE EXPLORATION OF HYPOTHESES REGARDING POTENTIAL mechanisms by which HPV increases HIV infection risk in young women".

4. Discussion last paragraph lines 358-64 could be rephrased as follows to more accurately reflect the study design: "The study assessed genital inflammation and HPV in specimens representing the mucosal surface over which sexual transmission of both viruses takes place, and demonstrated a DIFFERENCE IN GENITAL CYTOKINES BY HPV STATUS OVER TIME, WHICH INFORMS POSSIBLE MECHANISMS TO EXPLAIN THE COMPLEX epidemiological relationship between HPV and HIV risk. While these data do not rule out possible unmeasured explanations for the increased HIV risk associated with HPV infection, they do provide compelling biological evidence THAT the immune response IS ELEVATED IN HPV POSITIVE WOMEN, CONSISTENT WITH INCREASED HIV infection risk.

A few additional comments not previously highlighted for consideration:

1. The authors make a public health argument that HPV vaccination may reduce HIV risk, and support this with evidence that the associations are observed when restricting the HPV outcomes to the HPV-vaccine types. A more interesting analysis may be to look at the risk of HIV among vaccine protected types, and then among all non-vaccine protected types, relative to HPV negative women. This may give a better understanding of the excess risk of vaccine preventable vs. non-preventable types to the overall HPV risk, which in the presence of significant multiple infection may be more important when making claims about population attributable fraction (e.g., if we eliminated 9 types, how much HIV would we prevent given the risk remaining with the other 30+?).

2. A prior reviewer commented on not having cervical cytology. While I agree with the authors that this is unlikely to confound the analysis presented, their statement on line 346 (clean) that there would be little if any abnormal cytology in these young women is probably incorrect. Younger women usually have highest rates of abnormal cytology, though usually low grade. Just a point of clarification.

3. It might be considered, given the long interval between sampling for HPV detection, that an additional adjustment is made for the time between baseline HPV detection and HIV acquisition.

4. On page 6 lines 137-141 - the authors should consider the observation by Nowak et al. (JID 2011;203(8):1182-91) that multiple type HPV infections increase significantly immediately following HIV acquisition. Given the long sampling interval defining the HPV categories (clearance, persistence, acquisition), this 'reverse causality' should be considered in the context of the role of memory T-cell depletion during acute HIV infection.

5. In general, the use of the terms cleared vs. controlled is not just semantic when attempting to extrapolate the observed data (observed clearance) to biological mechanism (possible that observed clearance biologically reflects immune control of virus). Similarly, acquisition may reflect reactivation which is immunologically mediated. So while it is understood that there is no way to differentiate new infection from acquisition or cleared infection from controlled infection, the biological inferences would be quite different.

Response to Referees

Reviewer #5, expert in HPV epidemiology (Remarks to the Author):

“This is an important descriptive study which lends important biological marker data to help generate specific mechanistic hypotheses for the consistent epidemiologic associations between HPV infection and increased HIV infection risk.

The authors have generally responded well to the comments of the reviewers. In my view, the disconnect between the reviewers and the authors is the fundamental tension between how far epidemiologic research can contribute to demonstration of causality when compared with experimental laboratory studies, remaining cognizant that the laboratory evidence may be so highly contrived that the extrapolation of results to a complex human biological system may also be limited.

To that end, in this reviewer's opinion, the study was well-designed epidemiologically and used state-of-the-art biomarker measures both for HIV, HPV, and cytokines. These are all commonly used in epidemiologic research.

It is suggested that the authors carefully review the manuscript for all causal inference statements, and tone them down to better reflect the observational nature of what is really a hypothesis-generating dataset (albeit a very important one). A few possible (non-exhaustive) examples:

1. Title could be changed to: "Differences in genital cytokine milieu by HPV infection status in women at high risk of HIV acquisition". This title does not implicitly assume that the differences are triggered by HPV infection, but may be correlatively associated with other changes in the female genital tract (FGT). There are valid hypotheses to suggest this as outlined in Nowak et al (JID 2011;203(8):1182-91) and Averbach (as referenced by the authors)

We thank the reviewer for their valuable insight. The title was changed to capture the sentiment of the suggestion: “HPV infection and the genital cytokine milieu in women at high risk of HIV acquisition”

2. Discussion first sentence: "This study tested the hypothesis that HPV can increase HIV acquisition risk by modulating the female genital mucosa in a way that may facilitate HIV infection". In fact, this is not what the study tested. The study tested the hypothesis that the FGT cytokine milieu would differ by HPV status, and the nature of the difference could lead to more refined hypotheses regarding a biological mechanism to explain the increased risk of HIV observed in HPV positive women (especially those with clearance).

The reviewer's point is well taken. We have changed the first paragraph of the discussion to relay the suggested changes: “This study confirmed the epidemiological link between HPV and increased risk of HIV infection, and tested the hypothesis that the female genital tract cytokine milieu would differ by HPV status. The data presented are in support of this hypothesis, particularly in women who clear their HPV infection. Since clearance was most consistently associated with increased HIV risk, and with the broadest

cytokine response, including all cytokines previously associated with HIV risk in this cohort⁴¹, the data underscore the importance of further investigation to fully understand the complex immunological environment that may mediate the increased risk of HIV associated with HPV infection.”

3. Discussion last paragraph line 355 (clean): add the following (CAPS) to sentence: "In summary, we provide here DATA HELPFUL IN THE EXPLORATION OF HYPOTHESES REGARDING POTENTIAL mechanisms by which HPV increases HIV infection risk in young women".

The sentence is now expanded to acknowledge the hypothesis-generating nature of the work and reads as: In summary, although not a definitive mechanism, these data provide some insight into how HPV might increase HIV risk and highlight the importance of further studies to validate and expand on the concepts raised. The data suggest, among others, an urgency in understanding and controlling the immune predictors of HPV status and multiple type infection, in both HIV infected and uninfected women; in determining the cellular or other factors associated with the cytokines observed in the different HPV status categories; and in understanding the immune impact of HPV infection in vaccinated individuals and its relation to HIV risk.

4. Discussion last paragraph lines 358-64 could be rephrased as follows to more accurately reflect the study design: "The study assessed genital inflammation and HPV in specimens representing the mucosal surface over which sexual transmission of both viruses takes place, and demonstrated a DIFFERENCE IN GENITAL CYTOKINES BY HPV STATUS OVER TIME, WHICH INFORMS POSSIBLE MECHANISMS TO EXPLAIN THE COMPLEX epidemiological relationship between HPV and HIV risk. While these data do not rule out possible unmeasured explanations for the increased HIV risk associated with HPV infection, they do provide compelling biological evidence THAT the immune response IS ELEVATED IN HPV POSITIVE WOMEN, CONSISTENT WITH INCREASED HIV infection risk.

We thank the reviewer for their suggestion and note that these changes were made. The section now reads as: The study assessed genital inflammation and HPV in specimens representing the mucosal surface over which sexual transmission of both viruses takes place, and demonstrated a difference in genital cytokines by HPV status over time, which informs possible mechanisms to explain the complex epidemiological relationship between HPV and increased HIV risk. While these data do not rule out possible unmeasured explanations for the increased HIV risk associated with HPV infection, they do provide compelling biological evidence for elevated immune responses in HPV positive women, consistent with increased HIV infection risk.

We thank the reviewer for their thorough review of our work and for the constructive responses. We have reviewed our manuscript and refined (toned down) statements that claim demonstration of causality. In addition to the above suggestions, the following changes were made:

Line 27: Running title: **Mucosal cytokines by HPV status**

Lines 80 – 84: **Here we hypothesized that the female genital immune environment may differ according to HPV infection status; a concept that, if confirmed, would shed light on possible mechanisms for the reported associations between HPV infection and HIV risk. It is biologically plausible for cervical HPV infection, or other potentially correlated changes in the female genital tract to impact HIV risk, since sexual transmission of HIV is primarily mediated through interaction of the virus with genital cellular targets for infection, in the same mucosa where HPV replicates.**

Lines 86 – 88: **It is therefore reasonable that immune cell recruitment to eliminate or contain HPV infection in the genital epithelium may promote an immune environment that favors HIV infection.**

Lines 185 – 188: **These data suggest that the high frequency of HPV clearance in the cohort underscores its large contribution to increased rates of HIV acquisition, and support the hypothesis that effective immune responses against HPV infection may contribute to HIV risk^{19,21,39}.**

Lines 191 – 192: **Given the associations between HPV and HIV observed in this study, we hypothesized that the mucosal cytokine milieu may vary by HPV status.**

Lines 199 – 203: **Cytokines previously associated with HIV risk in CAPRISA 004 participants⁴¹ (IL-8, MIP-1 α , and MIP- β) were elevated in the genital tract of HPV positive women relative to HPV negative women (Figure 3). A similar profile of elevated cytokine concentrations was also observed on infection with any of the 9 vaccine-preventable HPV types relative to HPV negative women (Figure 3b).**

A few additional comments not previously highlighted for consideration:

1. The authors make a public health argument that HPV vaccination may reduce HIV risk, and support this with evidence that the associations are observed when restricting the HPV outcomes to the HPV-vaccine types. A more interesting analysis may be to look at the risk of HIV among vaccine protected types, and then among all non-vaccine protected types, relative to HPV negative women. This may give a better understanding of the excess risk of vaccine preventable vs. non-preventable types to the overall HPV risk, which in the presence of significant multiple infection may be more important when making claims about population attributable fraction (e.g., if we eliminated 9 types, how much HIV would we prevent given the risk remaining with the other 30+?).

We thank the reviewer for this insightful suggestion. We updated the results and discussion to reflect our additional analysis of the contribution that infection with nonavalent vaccine types, non-vaccine types, or no HPV has to the risk of HIV infection (Table 2). Assuming that immune responses associated with vaccine-targeted and cross-protected HPV types may be

important in these analyses, we assessed types included in the Gardasil®-9 vaccine, since this vaccine includes most of the HPV types that may receive cross-protection if using the other two vaccines. The data demonstrate a slightly higher HIV IR/100py in women with detectable Gardasil®-9 types 7.3 (95% CI 5.1 to 10.0) compared to women without (5.7, 95% CI 3.6 to 8.6), and only the risk associated with Gardasil®-9 type infection was significantly different to the HIV risk observed in HPV negative women. However, the HIV risk associated with infection by non-vaccine types exhibited a statistical trend (noting the decreased power for this analysis). Although our data demonstrate a potential for HPV vaccines to impact HIV risk, larger studies of HIV incidence in regions with differential HPV vaccine uptake are required to comprehensively determine the excess HIV risk attributed to infection with vaccine preventable versus non-preventable types. Further, considering our data on overlaps in cytokine profiles associated with HIV risk and with clearance, persistence and acquisition of HPV, additional studies are required to determine whether the impact of HPV vaccination on HIV risk may be bolstered through complementary efforts to control the potential immunological impact of infection with non-vaccine HPV types.

We have updated Table 2, the results text, and the discussion as follows:

Lines 136 – 138: Infection with non-Gardasil®-9 HPV types was associated with a 2-fold increase in HIV risk relative to HPV negative women, although not statistically significant (Table 2).

Lines 283 - 297: Considering the association between increased risk of HIV acquisition and infection by vaccine-preventable HPV types, these data underscore the importance of protecting young women from HPV infection through vaccination and cross-protection⁴⁶ against the broadest number of oncogenic HPV strains. The data also suggest the potential for increased HIV risk if infected by non-vaccine types, and emphasizes the importance of larger studies of HIV incidence in regions with differential HPV vaccine uptake to comprehensively determine the excess HIV risk attributed to infection with vaccine-preventable versus non-preventable types. Our finding that infection with any of the 9 vaccine-preventable HPV types was associated with raised concentrations of several genital cytokines associated with HIV risk suggests that any HIV protection provided by HPV vaccination may be biologically mediated. Considering the observed overlap in cytokine profiles associated with HIV risk and HPV clearance, persistence and acquisition, additional studies will be required to determine whether the ability of HPV vaccination to prevent HIV may need to be bolstered through complementary efforts to address any immunological effects associated with infection by non-vaccine HPV types.

2. A prior reviewer commented on not having cervical cytology. While I agree with the authors that this is unlikely to confound the analysis presented, their statement on line 346 (clean) that there would be little if any abnormal cytology in these young women is probably incorrect. Younger women usually have highest rates of abnormal cytology, though usually low grade. Just a point of clarification.

We thank the reviewer for the clarification. We specified that the absence of cervical cytology was unlikely due to confound the analyses presented since abnormal cytology in younger women would likely be low grade. Lines 370 – 374: Data on cervical cytology was not documented within the CAPRISA 004 trial. However, considering the population was HIV uninfected and mainly in their twenties, we would expect very little high grade abnormal cytology. However, given that this was a microbicide safety trial, extensive vaginal observation data were available, and these data do not support any widespread clinically obvious abnormal cytology likely to confound the analysis presented.

3. It might be considered, given the long interval between sampling for HPV detection, that an additional adjustment is made for the time between baseline HPV detection and HIV acquisition.

The survival analyses conducted here already take into account the time between baseline HPV detection and HIV acquisition. However, since HPV genotyping was only conducted 3 months into the study we considered an additional adjustment of reporting person-years from first HPV detection instead of from study start. We opted against excluding the first 3 months in the study in order to avoid over-estimation of HIV incidence. Nevertheless, the expected increases with this adjustment would have been marginal, with the risk of HIV infection on prevalent HPV infection being maintained at 2.5-fold relative to HPV negative women, suggesting that the additional adjustment would not drastically alter our findings of an association between HPV detection and HIV acquisition. Our methods section reflects the timing of HPV measurements and that the time at risk was defined from randomization.

4. On page 6 lines 137-141 - the authors should consider the observation by Nowak et al. (JID 2011;203(8):1182-91) that multiple type HPV infections increase significantly immediately following HIV acquisition. Given the long sampling interval defining the HPV categories (clearance, persistence, acquisition), this 'reverse causality' should be considered in the context of the role of memory T-cell depletion during acute HIV infection.

The reviewer makes an important point that has now been noted in our discussion. The Nowak paper very elegantly describes a relationship between HIV infection and increased numbers of detectable HPV (with, presumably, a greater propensity for acquisition, clearance, and/or persistence). The strength of the CAPRISA 004 study design lies in the emphasis on identifying the timing of HIV infections (Abdool Karim et al., 2010). At screening and at each monthly visit participants were tested for HIV with two rapid tests. Participants with two negative rapid tests continued follow-up in the study. If either of the tests were positive or indeterminate, then the participant was considered a suspected seroconverter and RNA PCR testing was performed to confirm HIV status. If the RNA PCR test was positive, Western blots and ELISAs were performed retrospectively on stored specimens to provide additional confirmatory information on the presence / absence of infection. The timing of HIV infection

was defined as the midpoint between last negative and first positive tests (by PCR or rapid HIV antibody test); and the date of seronegative infections was defined as 14 days before a positive PCR. While cytokine data were available for the majority of visits for each participant, HPV genotyping data were only available for baseline and study exit for most participants. In the cytokine analysis, considering the established relationship between HIV infection and immune dysregulation, our linear mixed models only included cytokine data collected from confirmed HIV-negative samples with complementary HPV genotyping data for that visit, and therefore reflects the genital mucosal immune milieu associated with a current HPV status in HIV negative women. The survival analysis, however, is indeed limited by a potential overestimation of the HIV incidence associated with multiple type infection, or with clearance, persistence, or acquisition of HPV. Because HPV genotyping data were available for baseline and study exit visits of most seroconverters, the longitudinal HPV classification would have been based on genotyping conducted after HIV infection. Since our study only included cytokine data for confirmed HIV negative specimens with matching HPV typing, we can only speculate on the immune profile associated with HIV infection that could promote an HPV status and larger studies designed to address this important question are sorely needed.

We updated the methods section for clarity on the timing of sampling and sample availability for the analyses; Lines 605 - 609: Cytokine data were available for the majority of visits for each participant, and specimens available from confirmed HIV-uninfected genital mucosal sampling visits were included in the cytokine analyses. HPV genotyping data were only available for baseline and study exit for most participants, and longitudinal HPV classification may have been based on genotyping conducted after HIV infection for some participants.

We also updated the discussion to emphasize the limitation in the estimation of HIV incidence by HPV status in the longitudinal classifications, and highlighted the importance of future studies to expand on the concepts raised in this paper, such as defining the immune predictors of HPV status and multiple type infection, in both HIV infected and uninfected women; determining the cellular or other factors that may be associated with the cytokines we observed in the different HPV status categories that may better describe a mechanism for the relationship between HPV and increased HIV risk in young women. Lines 345 - 363 now read as: Considering that acute HIV infection is demonstrated to increase the number of HPV types detected³⁹, the survival analysis is limited by this potential for overestimation of the HIV incidence associated with multiple type infection, or with clearance, persistence, or acquisition of HPV. Although the CAPRISA 004 trial included frequent HIV testing and rigorous retrospective viral load confirmations³⁷, HPV genotyping data were available for only baseline and study exit visits of most seroconverters, and longitudinal HPV classification would have been based on genotyping conducted after HIV infection. Additionally, the difficulty in timing the changes in HPV status in the context of HIV infection, we were unable to determine the degree of HPV-HIV co-infection occurring from the same infected partner. Cytokine data were, however, available for the majority of

visits for each participant. Considering the established relationship between HIV infection and immune dysregulation, our linear mixed models only included cytokine data collected from confirmed HIV-negative samples with complementary HPV genotyping data for that visit, and therefore reflects the genital mucosal immune milieu associated with a current HPV status in HIV negative women. Since cytokine data were available for confirmed HIV negative specimens, we can only speculate on the immune profile associated with HIV infection that could promote an HPV status. Our data does, however, support the hypothesis that immune profiles observed in HPV infection may contribute in some way to the established link between HPV infection and HIV risk.

5. In general, the use of the terms cleared vs. controlled is not just semantic when attempting to extrapolate the observed data (observed clearance) to biological mechanism (possible that observed clearance biologically reflects immune control of virus). Similarly, acquisition may reflect reactivation which is immunologically mediated. So while it is understood that there is no way to differentiate new infection from acquisition or cleared infection from controlled infection, the biological inferences would be quite different.

The manuscript now acknowledges the difficulty in these classifications. Lines 154 - 157 now include: Similarly, the term “acquired” here refers to the absence of DNA for an HPV type at one visit but the detection of it at the next consecutive visit, acknowledging that this definition may reflect new or re-infection, or reactivation³⁹. The discussion reiterates the acknowledgement in lines 341 - 345: The misclassification of HPV categories also extends to the difficulty in determining whether acquisition events represent reactivation or new infection, whether cleared infections refer to latent or eliminated infection, and the challenges of understanding the immunological profiles associated with each.

REVIEWERS' COMMENTS:

Reviewer #5 (Remarks to the Author):

The author's response was satisfactory - one clarification may be helpful

Infection with non-Gardasil®-9 HPV types

140 was associated with a 2-fold increase in HIV risk relative to HPV negative women, 141 although not statistically significant (Table 2).

For this sentence, could the authors clarify whether this category included women with exclusively non-Gardasil-9 types (vs. co-infected with Gardasil-9 types)?

If this category does not include gardasil-9 types, it certainly suggests the risk to HIV transmission may not be related to the oncogenicity of the HPV, but to more general immune control of HPV. If this is true, then it would be helpful to state in the manuscript to the % of prevalent gardasil-9-type-infected women were co-infected with non-gardasil-9 types

RESPONSE TO REVIEWERS' COMMENTS:

Reviewer #5 (Remarks to the Author):

The author's response was satisfactory - one clarification may be helpful

Infection with non-Gardasil®-9 HPV types
140 was associated with a 2-fold increase in HIV risk relative to HPV negative women,
141 although not statistically significant (Table 2).

For this sentence, could the authors clarify whether this category included women with exclusively non-Gardasil-9 types (vs. co-infected with Gardasil-9 types)?

If this category does not include gardasil-9 types, it certainly suggests the risk to HIV transmission may not be related to the oncogenicity of the HPV, but to more general immune control of HPV. If this is true, then it would be helpful to state in the manuscript to the % of prevalent gardasil-9-type-infected women were co-infected with non-gardasil-9 types

The participants in this group were indeed infected exclusively with non-Gardasil-9 HPV types. The percentage of women with prevalent Gardasil®-9 HPV who were co-infected with non-Gardasil®-9 types, 73%, is noted in the methods section, and the discussion is updated to further highlight the potential for HIV transmission to be related not only to oncogenicity but to general immune control of HPV infection. Although our data links oncogenicity and infection with vaccine-type HPV with HIV risk, we nonetheless emphasize the importance of larger studies of HIV incidence in regions with differential HPV vaccine uptake to comprehensively determine the excess HIV risk attributed to infection with vaccine-preventable versus non-preventable types.

Lines 662-663: Several baseline HPV infection categories were defined in order to assess the relationship between HIV acquisition and HPV DNA detection (prevalent HPV infection), multitype HPV infection, oncogenicity, and with vaccine type HPV infection. Considering the high number of multitype HPV infections, the potential for co-infection is particularly acknowledged within the latter two groups. For example, while women in the non-Gardasil®9 HPV types group were infected exclusively with non-Gardasil®9 HPV types, women in the Gardasil®9 group could be infected with both vaccine and non-Gardasil®9 HPV types, but had to be infected with at least one of the HPV types in the Gardasil®9 vaccine. Co-infection with non-Gardasil®9 HPV types was observed in 73% of the women in the Gardasil®9 HPV types group.

Lines 395-398: The data also suggest the potential for increased HIV risk if infected by non-vaccine types, indicating that the risk to HIV transmission may not be related to the oncogenicity of the HPV alone, but to general immune control of HPV. This concept emphasizes the importance of larger studies of HIV incidence in regions with differential HPV vaccine uptake to

comprehensively determine the excess HIV risk attributed to infection with vaccine-preventable versus non-preventable types.

RESPONSES TO EDITORIAL REQUESTS:

Your manuscript has been checked for clarity and against journal policies and formatting style. The issues listed below must be addressed. If using Microsoft Word, please use the tracked changes feature to make these changes.

*****Please provide a point-by-point response to the list below with your submission.*****

TITLE PAGE:

* I have made some edits to the abstract in the attached Word file. Please check that you agree with them.

We agree with the changes and have addressed the queries posed in the editorial comments.

With regard to the comment referring to the sentence “Among 48 vaginal cytokines profiled, cytokines associated with HPV prevalence, incidence, persistence and particularly clearance overlap substantially with...,” i.e. “Can you describe this in simpler terms? Would it be correct to collectively describe this group as HPV positive?”, the sentence was altered to read as: “Among 48 vaginal cytokines profiled, cytokines associated with HPV infection overlap substantially with...” This term encompasses all HPV infection categories, even HPV clearance, which could not be referred to under the proposed term “HPV positive”.

With regard to the queries related to the inclusion of specific cytokine names, “It is unclear if these three cytokines are associated with HIV as well as HPV, or just with HIV, or something in between. Please either state cytokines for all groups, or do not specify the cytokines for any group,” we agree with the concern and have opted to remove the cytokine names. The sentence now reads as: “Among 48 vaginal cytokines profiled, cytokines associated with HPV infection overlap substantially with cytokines associated with HIV risk, but are distinct from those observed in HPV negative women.

* The abstract should briefly discuss the background and context of the work, followed by the major results and conclusions of the paper. The discussion of the current work should begin with "Here we report" or an equivalent phrase.

The abstract now follows this format.

* When discussing the current work in the abstract, please use the present tense.

The abstract is now written in the present tense.

MAIN TEXT:

* Please use the present tense when discussing the current work in the Introduction.

This has been addressed throughout the introduction.

*Please provide the actual cytokine concentrations and the calculated fold change in the Source data.

These have now been provided in the Source Data.

* Please explain in the Methods how the b-coefficient was calculated.

This has now been explained in the methods section (lines 710-719).

*What does the dotted line represent in the figures?

The relevant figures now include a description of the relevance of the dotted line: “The models test the hypothesis that the mean cytokine levels in comparator groups are equal i.e. $\beta=0$. The dotted line at $\beta=0$ distinguishes higher (to the right of the line) from lower (to the left of the line) mean cytokine differences in the respective categories relative to the remained HPV negative group.”

LANGUAGE AND STYLE:

* Please ensure that +/- values are defined at the first point of use within the text and figure legends and numbers of replicates are given.

Line 160: CD4+CCR5+ is explained in the text as: “...the T cells are relevant to HIV infection as preferred targets of infection (CD4 T cells expressing the CCR5 co-receptor for HIV entry i.e. CD4+CCR5+ T cells)

Line 180: “HPV DNA was detected in 73.8% (95% CI 70.7 to 76.9%) of participants at baseline (HPV+; Table 1).”

Table 1: In legend: “HPV+: cervicovaginal HPV DNA detected; HPV-: no cervicovaginal HPV DNA detected;”

* Wherever p-values are stated in the text and figure legends, please also state the name of the statistical test.

This change has been conducted throughout the manuscript.

METHODS AND DATA:

* Please provide the absolute cytokine concentrations as Supplementary Data, and describe in the Methods how the values shown in the figures were derived from these data. Please state how many times each sample was measured, with how many dilutions per sample.

The absolute cytokine concentrations and mean differences are now included as a Source document as requested in the in-text comment; and the methods section includes a description of how the values shown in the figures were derived from these data (lines 647-656; 710-719).

To address how many times each sample was measured, and the dilutions per sample, the following lines were included:

Lines 647-656: All specimens were plated undiluted, and all laboratory assays were conducted blinded. Duplicates were included within plates to assess intra-plate variability (n=20 wells duplicated per plate), and replicates were placed across separate plates to assess inter-plate variability (n=32 wells replicated across plates). Cytokine assessments were repeated if correlation coefficients of associations between replicates were below Spearman rho = 0.8, and if significant differences in magnitudes among replicates were observed. Individual cytokines undetectable in >40% of specimens were assessed as categorical values.

Line 623: “The concentrations of cytokines were previously measured in undiluted CVL fluid of CAPRISA 004 participants using Luminex multiplex assays²⁴.”

* In the Methods section, please provide sufficient information such that the experiments could reasonably be reproduced without reference to other papers, and avoid use of the term "as described previously".

Separate expanded paragraphs were inserted in the methods to describe the assessment of other sexually transmitted infections (lines 599-609), and the measurement of tenofovir concentrations in cervicovaginal fluid (lines 614-620).

* Please ensure that you have cited the trial registration number from ClinicalTrials.gov or an equivalent agency in the Methods section of the manuscript. If the trial is a phase 2 or 3 randomised control trial, please also ensure that you have provided the CONSORT checklist with your submission.

The trial registration number (NCT00441298) is provided in the methods section (line 529), and a consort diagram is included in the paper. The CONSORT checklist is now included in our resubmission.

* Please confirm that you have complied with all relevant ethical regulations for work with human participants, and that informed consent was obtained. Please state this in the Methods section, including the name of the board and institution that approved the study protocol.

Confirmed. These are addressed in the Methods section as follows:

Line 526-532: Added: "All women provided informed consent for participation in the trial, and participants were only included in this study if consenting to specimen storage for assessment of the trial's ancillary objectives (Figure 1). The trial (NCT00441298) and this secondary analysis was approved by the University of KwaZulu-Natal's Biomedical Research Ethics Committee (E111/06), Family Health International's Protection of Human Subjects Committee (#9946) and the South African Medicines Control Council (#20060835)."

* All *Nature Communications* manuscripts must include a section titled "Data Availability" as a separate section after the Methods section and before the References. For more information on this policy, and a list of examples, please see <http://www.nature.com/authors/policies/data/data-availability-statements-data-citations.pdf>

DISPLAY ITEMS:

* Data in tables must be free from bold/italic formatting unless this has been clearly defined in the footnote. Tables need to be black and white, fit onto a single A4 portrait page and can contain only one row of column titles. Finally, we are unable to merge cells or include vertical dividing lines or diagonal lines. Please format your tables accordingly.

Bold and italic removed. Headings underlined instead. Table 2 cannot be accommodated in a full page portrait and is drafted as a full page landscape.

* We allow up to 10 display items (figures or tables) in the main text. Please consider whether supplementary items could be transferred to the main manuscript to aid the

reader. In particular, we recommend that the Supplementary Figure 1 is made a main display items.

Our previous “Supplementary figure 1” (consort diagram) is now Figure 1. All other Figure titles have been updated.

* Please define any new abbreviations, symbols or colours present in your figures in the associated legends. Please do not use symbols in your legend, instead please write out the symbols in words (blue circles, red dashed line, etc.).

These were addressed.

* In each figure and supplementary figure where error bars are used, they must be defined. One statement at the end of each figure is sufficient if the error bars are equivalent throughout the figure.

The error bars depicted are described in the legends.

SUPPLEMENTARY INFORMATION:

* We do not edit Supplementary Information files; they will be uploaded with the published article as they are submitted with the final version of your manuscript. Any tracked changes should be removed from the file and the file should be provided as a PDF file. Supplementary Figures do not need to be provided separately.

Tracked changes have been removed. Changes include providing descriptions of abbreviations, and providing definitions for dotted lines on figures.

* Please supply legends for each Supplementary Movie/Audio/Data file in your cover letter (not in the Supplementary Information file). Please label each files as Supplementary Movie/Audio/Data 1, etc.

Included

* We encourage increased transparency in peer review by publishing the reviewer comments and author rebuttal letters of our research articles, if the authors agree. Such peer review material is made available as a supplementary peer review file. **Please state in the cover letter ‘I wish to participate in transparent peer review’ if you want to opt in, or ‘I do not wish to participate in transparent peer review’ if you don’t.** Failure to state your preference will result in delays in accepting your paper for publication.

Included in the cover letter

Please note: we allow redactions to authors’ rebuttal and reviewer comments in the interest of confidentiality. If you are concerned about the release of confidential data, please let us know specifically what information you would like to have removed. Please note that we cannot incorporate redactions for any other reasons. Reviewer names will be published in the peer review files if the reviewer signed the comments to authors, or if reviewers explicitly agree to release their name. For more information, please refer to our FAQ page at:

<https://www.nature.com/documents/ncomms-transparent-peer-review.pdf>

* An updated editorial policy checklist that verifies compliance with all required editorial policies must be completed and uploaded with the revised manuscript. All points on the policy checklist must be addressed; if needed, please revise your manuscript in response to these points. Please note that this form is a dynamic "smart pdf" and must therefore be downloaded and completed in Adobe Reader, instead of opening it in a web browser.

Editorial policy checklist: <https://www.nature.com/authors/policies/Policy.pdf>

* An updated reporting summary must be completed and uploaded with the revised manuscript. All points on the reporting summary must be addressed; if needed, please revise your manuscript in response to these points. Please note that this form is a dynamic "smart pdf" and must therefore be downloaded and completed in Adobe Reader, instead of opening it in a web browser.

Reporting summary: <https://www.nature.com/authors/policies/ReportingSummary.pdf>

* Your paper will be accompanied by a two-sentence Editor's summary, of between 250-300 characters including spaces, when it is published on our homepage. Could you please approve the draft summary below or provide us with a suitably edited version.

Cervicovaginal inflammation and human papillomavirus (HPV) are separately associated with increased risk of HIV acquisition. Here the authors longitudinally profile 48 cervicovaginal cytokines and HPV status in a large observational HIV high-risk cohort, and show the same cytokines associate with HPV infection and HIV risk.

* As part of our efforts to communicate our content to a wider audience, we

endeavour to highlight papers published in *Nature Communications* on the journal's Twitter account (@NatureComms). If you would like us to mention authors, institutions or lab groups in these tweets, please provide the relevant twitter handles in your cover letter upon resubmission.

* If you opted into the journal hosting details of a preprint version of your manuscript via a link on our dedicated website (<https://nature-research-under-consideration.nature.com>), it will remain on this site while you are revising your manuscript, as we consider the file to remain active. Should you wish to remove these details, please email naturecommunications@nature.com indicating your manuscript number and the link on our website that was previously sent to you. Please see our pre-publicity policy at <http://www.nature.com/authors/policies/confidentiality.html> For more information, please refer to our FAQ page at <https://nature-research-under-consideration.nature.com/posts/19641-frequently-asked-questions>

* In recognition of the time and expertise our reviewers provide to *Nature Communications*'s editorial process, as of November, 2018, we formally acknowledge their contribution to the external peer review of articles published in the journal. All peer-reviewed content will carry an anonymous statement of peer reviewer acknowledgement, and for those reviewers who give their consent, we will publish their names alongside the published article. For more information, please refer to our FAQ page at <https://www.nature.com/documents/ncomms-reviewer-information.pdf>

OPEN ACCESS:

Nature Communications is a fully open access journal. Articles are made freely accessible on publication under a CC BY license (Creative Commons Attribution 4.0 International License). This license allows maximum dissemination and re-use of open access materials and is preferred by many research funding bodies.

For further information about article processing charges, open access funding, and advice and support from Nature Research, please visit <http://www.nature.com/ncomms/about/open-access>

SUBMISSION INFORMATION:

In order to accept your paper, we require the following:

- * A cover letter describing your response to our editorial requests.
- * A separate document detailing your point-by-point response to any issues raised by our referees (please include the referees' comments in this document).
- * The final version of your text as a Word or TeX/LaTeX file, with any tables prepared using the Table menu in Word or the table environment in TeX/LaTeX and using the 'track changes' feature in Word.

* The complete author list provided in the article file, which must match that given on our manuscript tracking system. The author list in the main article file will be used during typesetting of your article.

* Production-quality versions of all figures, supplied as separate files containing all panels. To ensure the swift processing of your paper please provide the highest quality, vector format, versions of your images (.ai, .eps, .psd) where available. Please see our brief guide to manuscript submission for further details on the figure formats we can accept. Text and labelling should be in a separate layer to enable editing during the production process. If vector files are not available then please supply the figures in whichever format they were compiled in and not saved as flat .jpeg or .TIFF files. Any chemical structures or schemes contained within figures should additionally be supplied as separate ChemDraw (.cdx) files. If your artwork contains any photographic images, please ensure these are at least 300 dpi.

Figures 2 and 6 are saved in the format they were made, .pptx. The rest of the figures as .eps. Please contact me if I can assist in improving their quality in any way.

To ensure that your figures are accessible to colour-blind readers, we encourage you to use alternative colour schemes. For example, rainbow colour scales may be replaced by single-colour intensity scales or greyscale, and red/green image overlays may be replaced with magenta/green. For reference an example of R-script colour blindness palettes can be found here <https://cran.r-project.org/web/packages/viridis/vignettes/intro-to-viridis.html>. Another example for Python can be found here: <http://matplotlib.org/cmoccean/>

* The final version of any Supplementary Information (figures, tables, notes etc) in one PDF file. Please add a cover page to the Supplementary Information PDF, including the title of the manuscript and the first author's surname in the format 'Smith et al.' Please submit movies, audio files and data sets as separate files. See <http://www.nature.com/ncomms/submit/how-to-submit#Supplementary-information> for acceptable file formats/sizes.

** Please note that Supplementary Information must be finalised prior to acceptance of the paper. **

* If you wish, an interesting image (but not an illustration or schematic) for consideration as a 'Featured Image' on the Nature Communications homepage. Examples can be seen on our Facebook page: <http://go.nature.com/PGPizM> The file should be 1400x400 pixels in RGB format and should be uploaded as 'Related Manuscript File'. In addition to our home page, we may also use this image (with credit) in other journal-specific promotional material.

* A completed author checklist, uploaded as a Related Manuscript file type, available at:

<https://www.nature.com/documents/ncomms-manuscript-checklist.pdf>

* Completed and signed copies of our Multimedia License to Publish (LTP) for any

Featured Image suggestions (please use one form for each image and give a scientific description of the image in the 'title' field; do not use "Featured Image" as a title):

Multimedia Licence to Publish form

At acceptance, the corresponding author will be required to complete an Open Access Licence to Publish on behalf of all authors, declare that all required third party permissions have been obtained and provide billing information in order to pay the article-processing charge (APC) via credit card or invoice.

Please note that your paper cannot be sent for typesetting to our production team until we have received these pieces of information; **therefore, please ensure that you have this information ready when submitting the final version of your manuscript.** Springer Nature encourages all authors and reviewers to adopt an Open Researcher and Contributor Identifier (ORCID). ORCID is a community-based initiative that provides an open, non-proprietary and transparent registry of unique identifiers to help disambiguate research contributions. All authors who link their ORCID to their account in our submission system will have their ORCID published on their articles. Please note that this is only possible if ORCIDs are linked prior to acceptance, that is, it is not possible to add ORCIDs at proof.

Please ensure that all co-authors are aware that they can link their ORCIDs, so that it will display on this paper. If they so wish, they must do so before the paper is formally accepted. It will not be possible to add ORCIDs post-acceptance, e.g. at proof. To link an ORCID please follow these instructions:

1. From the home page of the MTS click on 'Modify my Springer Nature account' under 'General tasks'.
2. In the 'Personal profile' tab, click on 'ORCID Create/link an Open Researcher Contributor ID (ORCID)'. This will re-direct you to the ORCID website.
- 3a. If you already have an ORCID account, enter your ORCID email and password and click on 'Authorize' to link your ORCID with your account on the MTS.
- 3b. If you don't yet have an ORCID account, you can easily create one by providing the required information and then clicking on 'Authorize'. This will link your newly created ORCID with your account on the MTS.

If you experience problems in linking your ORCID, please contact Platform Support

We hope to hear from you within two weeks; please let us know if the process may take longer.

Best regards,

Tanya Bondar PhD
Senior Editor
Nature Communications

<http://orcid.org/0000-0003-4765-8910>

Nature Research journals encourage authors to share their step-by-step

experimental protocols on a protocol sharing platform of their choice. Nature Research's Protocol Exchange is a free-to-use and open resource for protocols; protocols deposited in Protocol Exchange are citable and can be linked from the published article. More details can found at www.nature.com/protocolexchange/about.